# RPA shields inherited DNA lesions for post-mitotic DNA synthesis

Aleksandra Lezaja[1], Andreas Panagopoulos[1,2], Yanlin Wen[1,2], Edison Carvalho [1], Ralph Imhof[1] & Matthias Altmeyer [1✉]

The paradigm that checkpoints halt cell cycle progression for genome repair has been challenged by the recent discovery of heritable DNA lesions escaping checkpoint control. How such inherited lesions affect genome function and integrity is not well understood. Here, we identify a new class of heritable DNA lesions, which is marked by replication protein A (RPA), a protein primarily known for shielding single-stranded DNA in S/G2. We demonstrate that post-mitotic RPA foci occur at low frequency during unperturbed cell cycle progression, originate from the previous cell cycle, and are exacerbated upon replication stress. RPA-marked inherited ssDNA lesions are found at telomeres, particularly of ALT-positive cancer cells. We reveal that RPA protects these replication remnants in G1 to allow for post-mitotic DNA synthesis (post-MiDAS). Given that ALT-positive cancer cells exhibit high levels of replication stress and telomere fragility, targeting post-MiDAS might be a new therapeutic opportunity.

[1] Department of Molecular Mechanisms of Disease, University of Zurich, Zurich, Switzerland. [2]These authors contributed equally: Andreas Panagopoulos, Yanlin Wen. ✉email: matthias.altmeyer@uzh.ch

Cell proliferation depends on the ability to transmit undamaged genetic material from one cell generation to the next. Cell cycle checkpoints play an important role in genome integrity maintenance across cell generations by slowing down or pausing cell cycle progression, thus allowing time for genome repair[1–4]. However, cell cycle checkpoints are not failsafe, and examples of imperfect checkpoint control and transmission of genomic lesions across cell cycle boundaries have emerged[5–7]. Checkpoint slippage is typically associated with endogenous DNA lesions, which over time and through multiple rounds of cell divisions can lead to an accumulation of mutations that interfere with cell functions and can drive cancer development[8]. Hence, although an infrequent event, checkpoint escape and inheritance of genomic lesions across cell generations has profound influence on human health and disease.

A major source of endogenous DNA damage comes from replication stress, which is exacerbated by oncogene activation during cancer development[9–13]. Intriguingly, in response to replication stress, 'unfinished business from S-phase' bypasses cell cycle checkpoint control[14]. Such checkpoint slippage may occur because under-replicated genomic regions resemble physiological replication intermediates (e.g. two converging replication forks), which stay below the radar of the DNA damage response and the cell cycle checkpoint machinery. In order to complete DNA replication at these under-replicated regions, cells employ mitotic DNA synthesis (MiDAS), a specialized replication program that uses replicative polymerases and break-induced replication (BIR) outside of S-phase[15,16]. To which extent MiDAS, in light of the brief duration of mitosis, is capable of completing genome duplication before sister chromatids separate and cells divide is an important yet unresolved question.

Remnants of incomplete replication can be inherited by the next cell generation, as revealed by replication stress-associated 53BP1 nuclear bodies that form upon exit from mitosis in G1 cells[17,18]. Sites marked by 53BP1 represent difficult-to-replicate regions, such as chromosomal breakpoints known as common fragile sites (CFS), and detection of 53BP1 nuclear bodies in G1 cells has become a widely used marker of heritable DNA lesions in many cancer and non-cancer cells and in patient-derived material[16–21].

Analogous to the cellular response to DNA double-strand breaks, 53BP1 recruitment to inherited DNA lesions depends on upstream chromatin modifications by ATM/MDC1 and RNF8/RNF168 and on multivalent chromatin binding by 53BP1[22–25]. Together with downstream effectors, 53BP1 shields inherited DNA lesions in G1 from unscheduled nucleolytic degradation in the absence of a replicated template DNA[26–30]. 53BP1-marked lesions are also associated with tuning G1 duration and thus with the decision whether and when cells commit to the next S-phase. Indeed, S-phase entry is significantly delayed or abolished in presence of elevated levels of inherited DNA lesions, suggesting that cell fate decisions at the G1/S transition are linked to genotoxic stress experienced in the previous cell cycle[31–33]. Moreover, 53BP1 nuclear bodies ensure proper replication timing to enable completion of genome duplication at inherited lesions and thereby safeguard genome stability[7].

The discovery of 53BP1 and its associated proteins as markers of heritable DNA lesions has spurred the notion that cell cycle checkpoints are laxer than originally thought, and that certain mechanisms of genome integrity maintenance surprisingly work across cell cycle boundaries rather than being confined to specific cell cycle phases. 53BP1 is now commonly employed as marker of heritable DNA lesions and replication stress-associated DNA damage, with prognostic value in clinically relevant contexts. However, considering the size and complexity of mammalian genomes and a growing list of intrinsically difficult to replicate genomic regions, whether 53BP1 serves as a universal marker for heritable lesions is unknown. Guided by the possibility that 53BP1 might only mark and protect a subset of heritable genomic lesions, we set out to test whether additional inherited DNA lesions exist and to characterize their features and implications for genome stability.

## Results

**RPA marks heritable DNA lesions distinct from 53BP1 nuclear bodies.** As the function of 53BP1 and its downstream effector proteins is to restrain resection of broken DNA by nucleases[34], and given that homology-directed repair is inhibited in the G1 phase of the cell cycle[35], we tested for the presence of single-stranded DNA (ssDNA) in newly born G1 cells. Replication protein A (RPA) binds to and protects ssDNA that occurs during DNA replication and upon DNA end resection in the context of homology-directed repair[36,37]. Congruously, its functions have been studied primarily in the S and G2 phase of the cell cycle. To assess heritable DNA lesions in a cell cycle resolved manner, we employed quantitative image-based cytometry (QIBC), an automated high-content microscopy approach that allows for image-based cell cycle staging of large cohorts of asynchronously growing cells[38,39]. We used this approach for its sensitivity and high throughput, which allows monitoring of thousands of cells per condition with the spatial resolution required to detect sub-micrometer-sized nuclear compartments, and for the possibility to analyze individual cells in specific phases of the cell cycle (using DNA content, EdU incorporation and/or nuclear Cyclin A levels as markers) without the need to synchronize cells and thereby perturb their normal proliferation program (Supplementary Fig. 1a).

By using QIBC, we readily detected 53BP1 nuclear bodies in G1 cells, and their number increased upon replication stress as expected (Fig. 1a). Both, a low concentration of the polymerase inhibitor aphidicolin (APH), or inhibition of the S-phase checkpoint kinase ATR (ATRi) resulted in elevated heritable DNA lesions marked by 53BP1 in G1, and combined treatment with APH and ATRi further enhanced this phenotype (Fig. 1a and Supplementary Fig. 1b, c). APH and ATRi synergize in this context, because low dose APH slows down DNA replication while ATRi destabilizes replication forks and pushes cells through S/G2 and into mitosis with unresolved replication intermediates[40–42].

Surprisingly, apart from 53BP1 nuclear bodies, we also readily detected RPA foci in G1 cells (Fig. 1b). Although lower in numbers, RPA foci in G1 were also induced upon replication stress treatments, very similar to 53BP1 (Fig. 1b and Supplementary Fig. 1d). The effect of APH was milder than of ATRi, yet increasing the APH concentration from 0.2 μM to 0.3 μM and to 0.4 μM led to an increase in RPA foci in G1, which was qualitatively very similar to the increase in 53BP1 foci in G1 (Supplementary Fig. 1e, f). At higher APH concentrations, the number of RPA and 53BP1 foci in G1 did not increase further, presumably because of delayed progression from S/G2 through mitosis and into G1. QIBC-based RPA foci detection in G1 cells was specific, as the signal was observed with two different antibodies against the RPA subunits RPA70 and RPA32 (Fig. 1b and Supplementary Fig. 1g, h), and because it was abolished upon RPA depletion (Supplementary Fig. 1i, j). RPA foci in G1 were also induced by a second ATRi (VE-821), but not by an inhibitor against the related DNA damage response kinase ATM (Supplementary Fig. 2a, b). As control, we verified that ATM inhibition suppressed γH2AX formation after ionization radiation (IR) as expected (Supplementary Fig. 2c). Importantly, while replication stress, and particularly ATRi, significantly enhanced

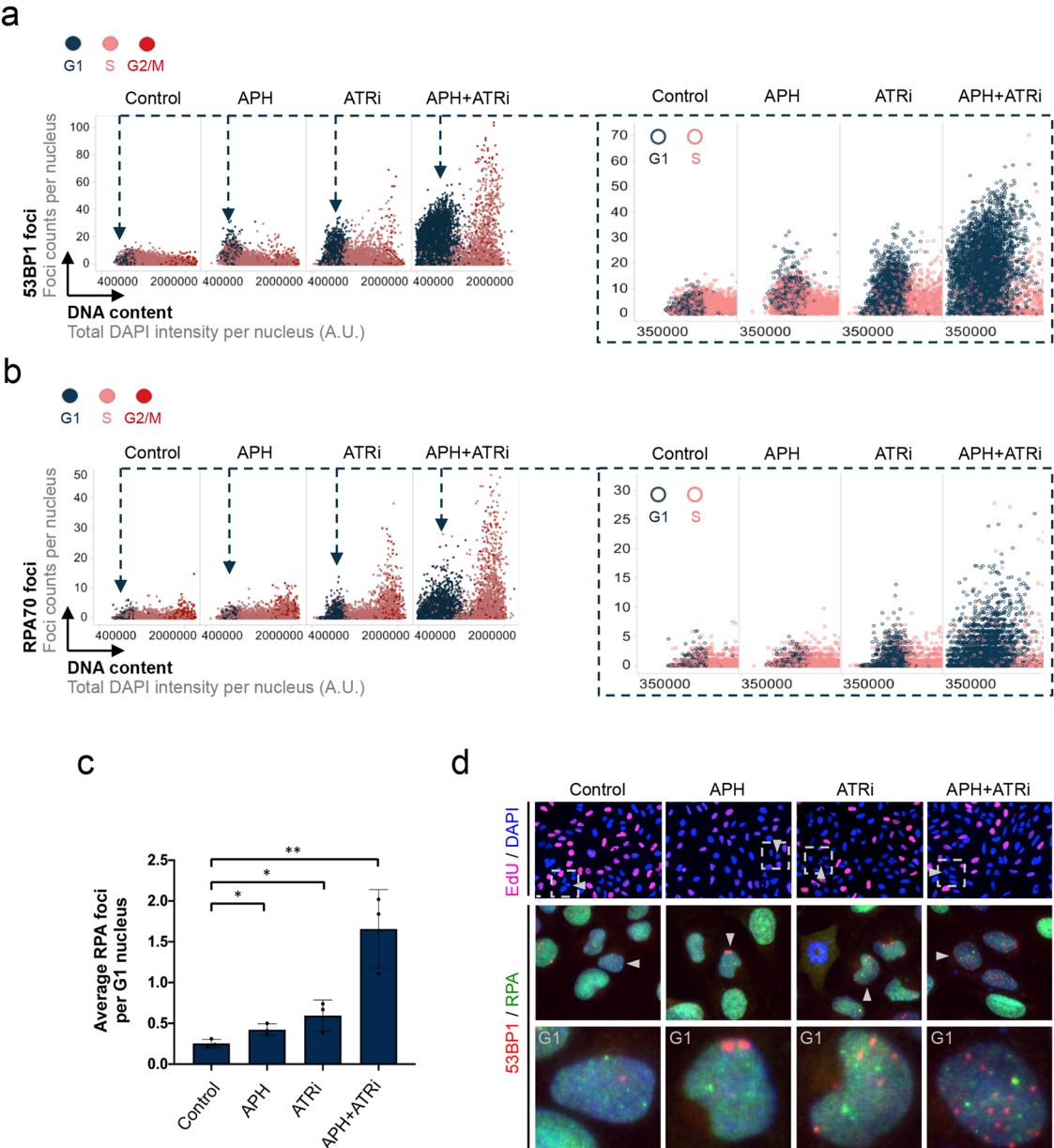

**Fig. 1 RPA marks heritable DNA lesions distinct from 53BP1 nuclear bodies. a** Asynchronously growing U-2 OS cells were treated with aphidicolin (APH, 0.2 μM), ATR inhibitor (ATRi, 1μM), or APH + ATRi for 24h as indicated. 5-Ethynyl-2'-deoxyuridine (EdU) was added 20 min before fixation. Cells were stained for 53BP1 and RPA70, and cell cycle staging was performed based on DAPI and EdU by QIBC (see Supplementary Fig. 1a and Methods section for details). Depicted are cell cycle-resolved scatter plots, in which G1 cells are labeled in blue, S phase cells in light red, and G2/M cells in dark red. Nuclear 53BP1 foci are on the y-axis, DNA content is on the x-axis. Each dot represents a single cell with its foci count. A rescaled version is shown on the right with a focus on cells in G1. Color-code as indicated. At least 1000 cells per condition were analyzed. **b** For the same cell populations depicted in (a) RPA foci were quantified and are shown in a cell cycle-resolved manner. **c** Average RPA foci counts in G1 in the different treatment conditions from $n = 3$ independent samples with $n_1 = 499$, $n_2 = 495$, $n_3 = 476$ (Control), $n_1 = 367$, $n_2 = 345$, $n_3 = 278$ (APH), $n_1 = 944$, $n_2 = 1046$, $n_3 = 968$ (ATRi), $n_1 = 959$, $n_2 = 1042$, $n_3 = 1044$ (APH + ATRi) cells in G1 per sample. Individual average values and means ± SD are shown. P-values were determined by two-tailed unpaired t-test; *$p < 0.05$ (exact p-values are $p = 0.0291$ and $p = 0.0392$, respectively), **$p < 0.01$ (exact p-value is $p = 0.0074$). **d** Representative images of individual cells in G1 are shown. The upper row shows the EdU and DAPI signals used by QIBC to identify G1 cells (see Supplementary Fig. 1a). The lower rows show G1 cells with 53BP1 and RPA foci. Scale bar: 10 μm. A. U., arbitrary units. Source data are provided as a Source Data file.

the formation of RPA foci in G1, even normally proliferating cells showed RPA foci in G1 in unchallenged condition (Fig. 1b–d).

When we investigated co-localization between 53BP1 and RPA foci in G1, we observed that these two markers showed largely distinct staining patterns, suggesting that they might mark different types of genomic lesions (Fig. 1d). To validate this finding, we used CRISPR/Cas9 to engineer stable 53BP1-GFP cells to express RPA70-mScarlet from the endogenous *RPA70* gene

locus. The mScarlet signal showed a typical RPA pattern, which overlapped with antibody-based RPA70 staining (Supplementary Fig. 3a), and which was reduced by siRNA against RPA70 (Supplementary Fig. 3b). By performing time-lapse microscopy with these cells, we observed discernible RPA foci forming in unchallenged conditions as cells exited mitosis. RPA foci formation preceded 53BP1 foci formation in G1, and rarely co-localized with 53BP1 (Supplementary Fig. 3c and Supplementary

Movie 1). Furthermore, RPA foci persisted longer than 53BP1 foci (Supplementary Movie 1) and could be followed through a complete cell cycle (Supplementary Fig. 3d and Supplementary Movie 2). Taken together, these results suggest that RPA-marked lesions are largely distinct from lesions marked by 53BP1. They are present at low frequency in newly born G1 cells during normal proliferation, and more frequently when cells experience replication stress.

**RPA-marked lesions in newly born G1 cells originate from the previous cell cycle and are associated with ssDNA.** APH and ATRi both affect the S-phase replication program. To test whether RPA foci observed in G1 cells in response to such treatments originate from the previous cell cycle, we repeated the experiment in absence or presence of CDKi RO-3306, which prevents entry into mitosis and cell division. Consistent with RPA foci in G1 being associated with replication stress during the previous cell cycle, CDKi abolished APH + ATRi-induced RPA foci in G1 (Supplementary Fig. 4a). Moreover, in time-course experiments, we observed that 2 h and 4 h of APH + ATRi treatment was sufficient to induce RPA foci in S and G2/M, but not in G1, whereas 8 h of treatment were needed to induce RPA foci in G1 (Supplementary Fig. 4b). Together these results suggest that the observed increase in RPA foci is due to problems during S/G2 of the previous cell cycle.

RPA is an ssDNA binding protein that protects replication and repair intermediates in S/G2, and we therefore wanted to test directly whether replication stress-associated ssDNA could also be detected in G1 cells. Native (non-denaturing) CldU (5-chloro-2-deoxyuridine) measurements by QIBC indeed revealed the presence of replication stress-associated ssDNA, with a cell cycle pattern resembling that of RPA foci (Supplementary Fig. 4c). Furthermore, S1 nuclease treatment abolished RPA foci in response to replication stress throughout the cell cycle, including RPA foci in G1 (Supplementary Fig. 4d). Replication stress thus not only interferes with replication completion during S/G2, but it is also associated with RPA-marked ssDNA lesions in daughter cells.

**RPA-marked inherited genomic lesions occur at telomeres.** In order to identify the genomic regions that give rise to RPA foci in G1 as a consequence of replication stress in the previous cell cycle, we turned to regions known for their intrinsic fragility. While common fragile sites (CFS) are associated mostly with 53BP1 nuclear bodies[17], early replicating fragile sites (ERFS), which break at the beginning of S-phase[43], seemed unlikely in light of our finding that 8 h of APH + ATRi was sufficient to induce RPA foci in G1 (Supplementary Fig. 4b). Telomeric regions at the ends of chromosomes represent a third class of fragile sites, which due to their highly repetitive, heterochromatic nature are difficult to replicate[44]. When we stained for telomeric-repeat binding factor 2 (TRF2) as marker of chromosome ends, we observed a clear co-localization between TRF2 and RPA-marked lesions in G1 cells (Fig. 2a and Supplementary Fig. 5a, b). Quantification of RPA foci in G1 and their co-localization with TRF2 yielded consistent results, with around 60–65% of all RPA foci found together with TRF2 (Fig. 2b–d). 53BP1, on the other hand, showed comparably poor co-localization with TRF2 (Supplementary Fig. 5c). Iterative indirect immunofluorescence imaging (4i)[45] and quantification of co-localization in G1 cells confirmed a higher degree of association between RPA and TRF2 as compared to 53BP1 and TRF2 (Supplementary Fig. 5d–f). Telomere FISH (fluorescent in situ hybridization) experiments with a telomere-specific probe validated the specificity of the TRF2 signal for chromosome ends (Supplementary Fig. 6a), and

that RPA foci in G1 cells co-localize with telomere sequences (Supplementary Fig. 6b). Furthermore, single plane confocal images of higher resolution confirmed that RPA foci in EdU-negative G1 cells co-localize with telomeres (Supplementary Fig. 6c). Thus, RPA-marked inherited genomic lesions in G1 are predominantly associated with telomeres.

**Telomeres acquire heritable ssDNA lesions on both strands.** To test whether replication stress-associated telomere lesions in G1 contain ssDNA regions, we performed telomere FISH experiments under native (non-denaturing) conditions in QIBC mode. Using a Tel C FISH probe, which binds to the complementary single-stranded G-rich strand of telomeres, this revealed increased telomeric ssDNA in response to replication stress, primarily in S/G2 but also in G1 (Fig. 2e and Supplementary Fig. 7a). Tel C foci in G1 cells co-localized with RPA foci, strongly indicating that RPA shields telomeric ssDNA in G1 (Fig. 2f and Supplementary Fig. 7b). Of note, Tel C foci co-localizing with RPA were also readily detected in unchallenged G1 cells, suggesting that while induced replication stress enhances heritable telomeric ssDNA formation, telomeric ssDNA also occurs due to endogenous replication problems.

Next, we employed a Tel G probe, which under non-denaturing conditions binds to the complementary single-stranded C-rich strand of telomeres. Also with this probe we observed replication stress-induced formation of ssDNA at telomeres (Supplementary Fig. 7c) with a similar pattern as obtained with the Tel C probe (Supplementary Fig. 7d). Consistently, the Tel G and the Tel C signals showed a clear positive correlation and co-localized (Supplementary Fig. 7e), and their induction by replication stress was dependent on the experiments being done under native, non-denaturing conditions (Supplementary Fig. 7e, f). These results suggest that ssDNA occurs on both strands of telomeric DNA in response to replication stress, and that RPA shields these inherited lesions in the subsequent G1 phase of the cell cycle.

**ALT-positive cancer cells are particularly prone to inherit telomere lesions from the previous cell cycle.** Telomere maintenance in cancer cells occurs by either of two mechanisms, by telomerase-mediated telomere elongation or by alternative lengthening of telomeres (ALT), a telomerase-independent mechanism that is active in around 10–15% of cancers[46–49]. U-2 OS cells are ALT-positive, raising the possibility that ALT might contribute to the formation of heritable ssDNA lesions upon replication stress. Prolonged exposure of U-2 OS cells to APH and ATRi causes telomere loss, as measured by telomere FISH on metaphase spreads (Supplementary Fig. 8a), and telomere fragility (Supplementary Fig. 8b). When we tested a second ALT-positive cell line, GM847, we could easily detect RPA foci in G1 by QIBC (in fact, much more clearly than in U-2 OS), and they co-localized with the telomeric protein TRF2 (Fig. 3a). We therefore compared two ALT-positive versus two ALT-negative cell lines by scoring the percentage of G1 cells positive for RPA foci. While the ALT-positive cell lines U-2 OS and GM847 showed up to 80% of G1 cells with RPA foci upon replication stress, the ALT-negative cell lines HeLa and RPE-1 showed RPA foci in G1 less frequently, with maximally 20% of positive G1 cells upon replication stress treatment (Fig. 3b–e and Supplementary Fig. 8c–e). In line, when we compared ALT-positive SW26 cells to telomerase-positive SW39 cells derived from the same IMR-90 cell line[50], we observed more pronounced RPA foci in G1 in SW26 (Fig. 3f, g and Supplementary Fig. 8f, g). Moreover, when we induced ALT in HeLa cells with long telomeres by depletion of the histone chaperone ASF1[51], RPA foci in

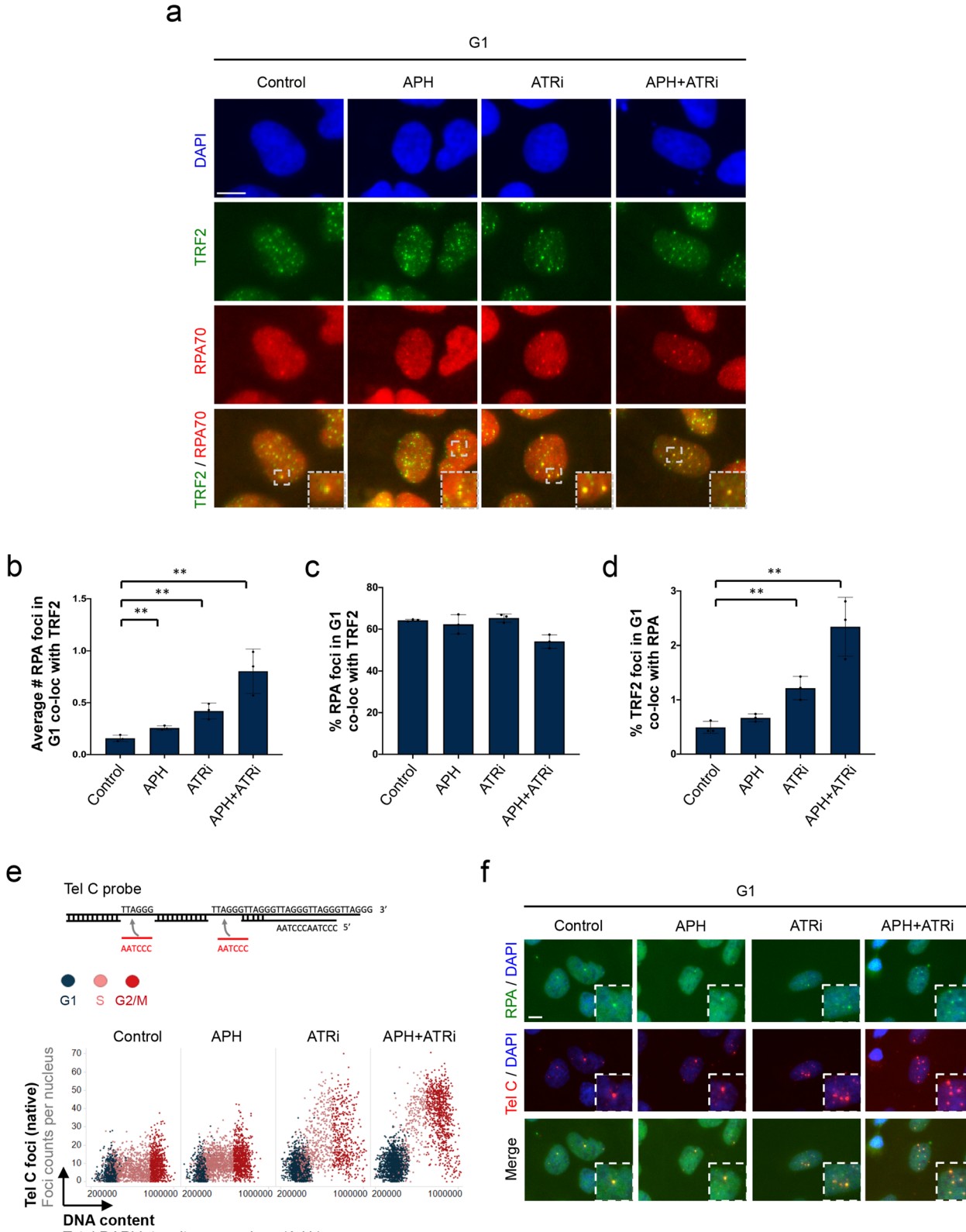

G1 cells increased (Supplementary Fig. 8h–j). As R-loops had been implicated in telomere maintenance by ALT[52], we depleted RNaseH1 in U-2 OS cells and observed elevated RPA foci in G1 (Supplementary Fig. 8k, l). Similarly, RPA foci in G1 were increased when RNaseH1 was depleted in GM847 cells (Supplementary Fig. 8m, n).

Telomere maintenance in ALT-positive cancer cells is thought to occur in ALT-associated PML bodies (APBs), specialized membraneless nuclear compartments and centers for telomere clustering[53–56]. Consistently, we observed that in ALT-positive cells RPA foci in G1 co-localized with PML bodies (Supplementary Fig. 8o), and that these co-localized with TRF2 (Supplementary

**Fig. 2 RPA-marked inherited genomic lesions occur at telomeres. a** Asynchronously growing U-2 OS cells were treated for 24h as indicated. Cells were stained for RPA and TRF2, and cell cycle staging to identify G1 cells was performed by QIBC. Representative images of individual G1 cells are shown. **b** Quantification of the average number of RPA foci co-localizing with TRF2 in G1 in different treatment conditions from $n = 3$ independent samples with $n_1 = 499$, $n_2 = 495$, $n_3 = 476$ (Control), $n_1 = 367$, $n_2 = 345$, $n_3 = 278$ (APH), $n_1 = 902$, $n_2 = 1046$, $n_3 = 968$ (ATRi), $n_1 = 959$, $n_2 = 1042$, $n_3 = 1044$ (APH + ATRi) cells in G1 per sample. Individual average values and means ± SD are show. P-values were determined by two-tailed unpaired t-test; **$p < 0.01$ (exact p-values are $p = 0.0094$, $p = 0.005$, and $p = 0.0066$, respectively). **c** Quantification of the percentage of RPA foci co-localizing with TRF2 in G1 corresponding to **b**. Individual average values and means ± SD are shown. **d** Quantification of the percentage of TRF2 foci co-localizing with RPA in G1 corresponding to **b**. Individual average values and means ± SD are shown. P-values were determined by two-tailed unpaired t-test; **$p < 0.01$ (exact p-values are $p = 0.0066$ and $p = 0.0044$, respectively). **e** Native (non-denaturing) FISH-IF to detect ssDNA at telomeres in U-2 OS cells. A telomeric FISH probe (Tel C) was used to detect single-stranded G-rich telomeric sequences. Tel C foci counts were quantified by QIBC and are plotted in a cell cycle-resolved manner. At least 1000 cells per condition were analyzed. **f** Representative images of G1 cells from native FISH-IF staining using the telomeric Tel C probe and RPA. Scale bars: 10 μm. A. U., arbitrary units. Source data are provided as a Source Data file.

Fig. 8p, q). Thus, replication stress-associated heritable ssDNA lesions occur in G1 on both strands of telomeric sequences, preferentially in ALT-positive cancer cells, and they are bound by RPA within APBs.

**RPA-marked lesions are primed for post-mitotic DNA synthesis (post-MiDAS).** Our findings suggest that telomere replication in ALT-positive cancer cells is inherently prone to cause heritable ssDNA lesions at telomeric DNA, which present as RPA foci in G1 cells. We next aimed to investigate the mechanism of telomere maintenance at heritable ssDNA, caused by replication stress during the previous cell cycle. In principle, gap filling DNA synthesis in G1 could provide cells with a means for transgenerational telomere maintenance. In other words, ssDNA lesions and potentially other replication stress-induced atypical DNA structures at telomeres could be dealt with consecutively as cells go from one round of the cell cycle to the next, without necessarily triggering cell cycle arrest or cell death. To investigate this possibility, we first elevated replication stress at telomeres by downregulation of the Fanconi anemia group protein FANCM, a telomere-associated factor that was recently identified to counteract telomeric replication stress specifically in ALT-positive cells[57–59]. FANCM depletion resulted in an increase in RPA foci, including enhanced RPA foci in cells in G1 that had escaped G2/M checkpoint surveillance (Fig. 4a–c). RPA foci in G1 co-localized with TRF2 (Fig. 4d, e), indicating that FANCM depletion indeed causes replication stress at telomeres, which then leads to ssDNA in the subsequent cell cycle. Previous work has shown that FANCM limits replication stress at telomeres in ALT-positive cells by restraining the activity of the BLM helicase[59]. Consistently, we could rescue the RPA foci in G1 that formed upon FANCM loss by BLM co-depletion (Supplementary Fig. 9a–c). As replication stress at telomeres has been linked to MiDAS[60,61], we then asked whether RPA foci in G1 require the MiDAS-associated factor RAD52. Indeed, knockdown of RAD52 reduced replication stress-induced RPA foci in G1 (Supplementary Fig. 9d, e). Next, we directly tested whether we could detect DNA synthesis in G1 cells, hypothesizing that such a mechanism might work as a backup for MiDAS. We used again asynchronously growing cells to avoid synchronization artifacts, and added the microtubule interfering agent nocodazole for 2 h to prevent mitotic cells from newly entering G1. In presence of nocodazole, we added EdU for the last 1 h to label newly synthesized DNA, and we used QIBC to identify G1 cells based on a 2 N DNA content and low EdU and/or low Cyclin A (Fig. 4f). This procedure allowed us to exclude DNA synthesis coming from MiDAS (as cells remained blocked from mitotic progression) and enabled us to specifically look at DNA synthesis occurring in G1. While it was difficult under standard conditions to detect EdU foci in G1 phase in control U-2 OS cells, FANCM-depleted cells showed clear EdU foci in G1, which co-localized with RPA and

with the telomere marker TRF2 (Fig. 4f). Consistent with our previous results, BLM co-depletion suppressed EdU incorporation at telomeres and RPA foci formation in G1 (Supplementary Fig. 9f, g).

When we increased the EdU concentration from 20 μM to 100 μM, we readily observed EdU foci in G1 cells also in naïve U-2 OS cells without FANCM depletion (Fig. 5a). Replication stress treatments increased the frequency of EdU foci in G1 (Fig. 5b), and 90–100% of EdU foci in G1 co-localized with RPA (Fig. 5c). To confirm the specificity of these signals, we tested whether inhibition of polymerases would abolish EdU foci in G1. Polymerase inhibition by a high dose of APH, or depletion of nucleotides by hydroxyurea (HU) blocked DNA replication in S-phase (Supplementary Fig. 10a), and also abolished EdU foci formation in G1, while leaving RPA foci in G1 unaffected (Fig. 5d and Supplementary Fig. 10b). In GM847 cells, which show pronounced RPA foci in G1 even in absence of exogenous replication stress (Fig. 3a), we also readily observed EdU foci in unchallenged G1 cells (Fig. 5e). Consistent with the U-2 OS data, EdU foci detected in GM847 G1 cells were abolished by co-incubation with APH during the EdU pulse, while RPA foci were not (Fig. 5e).

As MiDAS uses RAD52-mediated break-induced replication[62–64], we tested whether loss of RAD52 would affect EdU incorporation in G1 cells. In both RAD52-depleted and in RAD52 knockout cells (Supplementary Fig. 10c) EdU foci in G1 were reduced, although not completely abolished (Fig. 5f, g). Together with the finding that RAD52 foci co-localize frequently with RPA foci in G1 (Supplementary Fig. 10d, e), these data indicate a potential role of RAD52 for telomere DNA synthesis in G1.

In summary, we suggest that single-stranded replication intermediates occurring at ALT telomeres can be transmitted through mitosis and into the next cell cycle, where they are bound by RPA and undergo telomere maintenance involving DNA synthesis in G1. In analogy to mitotic DNA synthesis (MiDAS), we refer to this process as post-mitotic DNA synthesis (post-MiDAS).

**RPA protects ALT telomeres during post-MiDAS from breakage.** Last, we tested the relevance of RPA-mediated ssDNA shielding for ALT telomere stability during post-MiDAS in G1. As RPA is essential for DNA replication during S-phase, RPA depletion in asynchronous cells could cause indirect effects on G1 cells. We therefore sought to deplete RPA specifically in G1 cells, without interfering with the replication process itself. To this end, we arrested cells in G2/M by nocodazole, performed a mitotic shake-off and released the arrested cells into fresh medium for 4 h, allowing them to pass through mitosis. We then performed RPA depletion for 20 h in presence of CDKi RO-3306 (to block remaining S/G2 cells from cell division) and EdU (to mark cells entering S-phase and to be

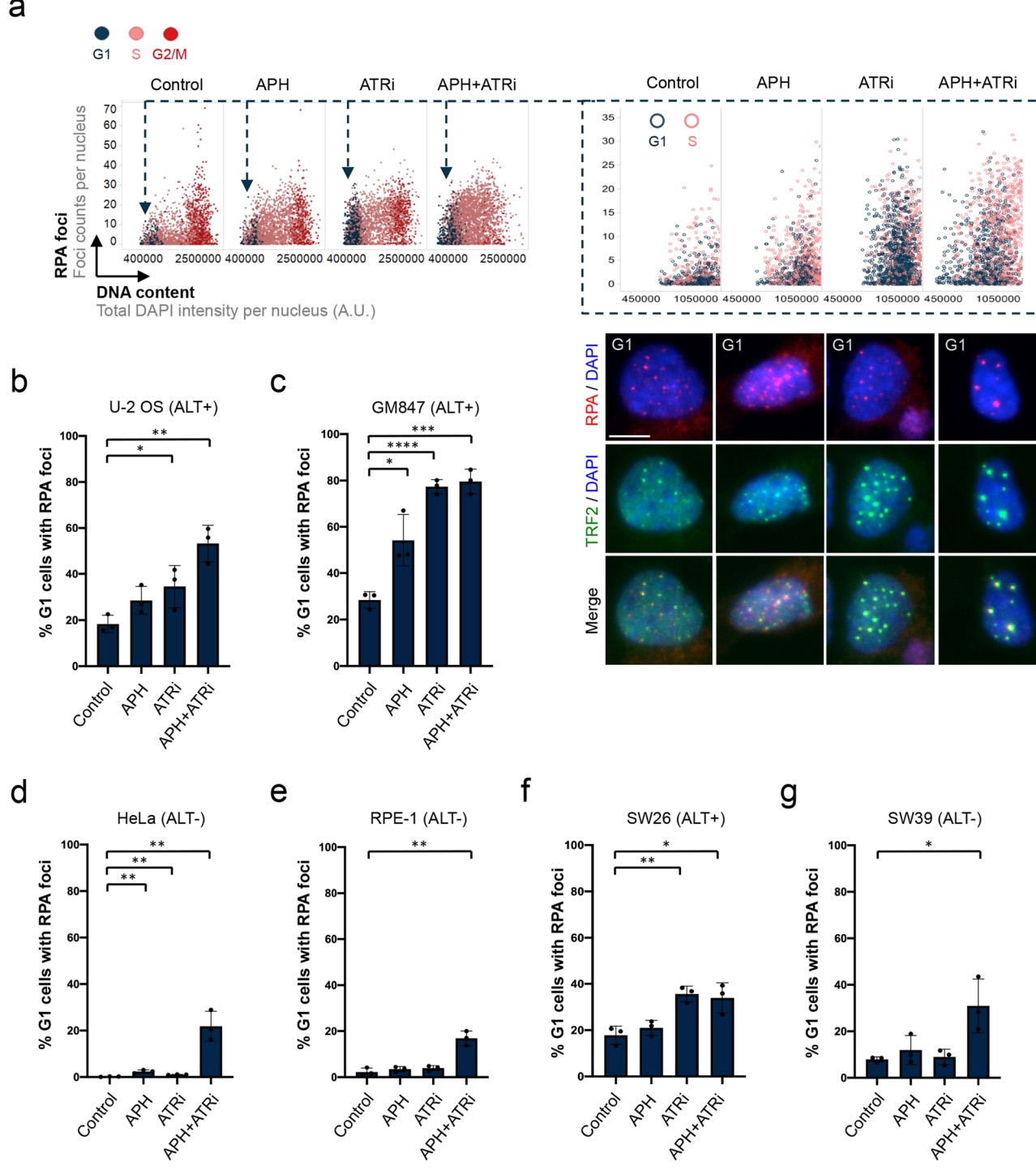

able to discriminate them from G1 cells), in order to assess telomere damage in G1 (Fig. 6a). RPA levels were reduced, although RPA loss was incomplete after only 20 h of knockdown (Supplementary Fig. 10f). On the other hand, a sizable number of G1 cells could be analyzed by this approach (Supplementary Fig. 10g). In G1 cells, we observed an increase in broken telomeres, as measured by TRF2 foci co-localizing with the DNA damage marker 53BP1 (Fig. 6b). In particular the percentage of G1 cells with three or more 53BP1-TRF2 co-localizations was elevated upon acute loss of RPA protection, suggesting more telomere breakage when RPA function was impaired (Fig. 6c, d). We conclude that RPA protects heritable ssDNA at ALT telomeres for post-MiDAS in G1 (Fig. 7).

## Discussion

As genome integrity maintenance is key to faithful propagation of genetic information from one cell generation to the next, cells employ checkpoints at multiple stages of cell cycle progression. For certain types of DNA lesions, tight checkpoint control is essential to maintain a stable genome. Chromosome breaks, for instance, if left unrepaired, give rise to acentric chromosomes and micronuclei, which can cause severe genome rearrangements[65]. Replication stress, on the other hand, involves a plethora of replication intermediates and genomic lesions, some of which can easily escape checkpoint surveillance, in particular in cancer[4]. Here, we show that replication intermediates at chromosome ends pass through mitosis and are shielded by RPA in the subsequent

**Fig. 3 ALT-positive cancer cells are particularly prone to form heritable telomere lesions. a** ALT-positive GM847 cells were treated as indicated and stained for EdU, RPA and TRF2. Cell cycle staging to identify G1 cells was performed by QIBC from at least 1000 cells per condition. Cell cycle-resolved scatter plots are depicted with RPA foci on the y-axis and DNA content on the x-axis. A rescaled version is shown on the right with a focus on cells in G1. Representative images of G1 cells are shown below. **b** Quantification of the percentage of ALT-positive U-2 OS cells with RPA-marked lesions in G1 in different treatment conditions from $n = 3$ independent samples with $n_1 = 499$, $n_2 = 495$, $n_3 = 476$ (Control), $n_1 = 367$, $n_2 = 345$, $n_3 = 278$ (APH), $n_1 = 944$, $n_2 = 1046$, $n_3 = 968$ (ATRi), $n_1 = 959$, $n_2 = 1042$, $n_3 = 1044$ (APH + ATRi) cells in G1 per sample. Individual values and means ± SD are shown. P-values were determined by two-tailed unpaired t-test; *$p < 0.05$ (exact p-value is $p = 0.0462$), **$p < 0.01$ (exact p-value is $p = 0.0023$). **c** Quantification of the percentage of ALT-positive GM847 cells with RPA-marked lesions in G1 in different treatment conditions from $n = 3$ independent samples with $n_1 = 323$, $n_2 = 221$, $n_3 = 306$ (Control), $n_1 = 235$, $n_2 = 221$, $n_3 = 206$ (APH), $n_1 = 407$, $n_2 = 442$, $n_3 = 437$ (ATRi), $n_1 = 247$, $n_2 = 287$, $n_3 = 241$ (APH + ATRi) cells in G1 per sample. Individual values and means ± SD are shown. P-values were determined by two-tailed unpaired t-test; *$p < 0.05$ (exact p-value is $p = 0.0186$), ***$p \leq 0.001$ (exact p-value is $p = 0.0002$), ****$p \leq 0.0001$. **d** Quantification of the percentage of ALT-negative HeLa cells with RPA-marked lesions in G1 in different treatment conditions from $n = 3$ independent samples with $n_1 = 1079$, $n_2 = 1090$, $n_3 = 1208$ (Control), $n_1 = 789$, $n_2 = 775$, $n_3 = 882$ (APH), $n_1 = 1192$, $n_2 = 1232$, $n_3 = 1242$ (ATRi), $n_1 = 260$, $n_2 = 313$, $n_3 = 225$ (APH + ATRi) cells in G1 per sample. Individual values and means ± SD are shown. P-values were determined by two-tailed unpaired t-test; **$p < 0.01$ (the exact p-values are $p = 0.0067$, $p = 0.0011$, and $p = 0.0045$, respectively). **e** Quantification of the percentage of ALT-negative RPE-1 cells with RPA-marked lesions in G1 in different treatment conditions from $n = 3$ independent samples with $n_1 = 1984$, $n_2 = 1761$, $n_3 = 2281$ (Control), $n_1 = 588$, $n_2 = 563$, $n_3 = 641$ (APH 0,4 μM), $n_1 = 2668$, $n_2 = 2434$, $n_3 = 2586$ (ATRi), $n_1 = 785$, $n_2 = 746$, $n_3 = 860$ (APH + ATRi) cells in $G_1$ per sample. Individual values and means ± SD are shown. P-values were determined by two-tailed unpaired t-test; **$p < 0.01$ (the exact p-value is $p = 0.0021$). **f** Quantification of the percentage of ALT-positive SW26 cells with RPA-marked lesions in G1 in different treatment conditions from $n = 3$ independent samples with $n_1 = 1091$, $n_2 = 781$, $n_3 = 929$ (Control), $n_1 = 737$, $n_2 = 594$, $n_3 = 642$ (APH), $n_1 = 806$, $n_2 = 770$, $n_3 = 706$ (ATRi), $n_1 = 501$, $n_2 = 355$, $n_3 = 396$ (APH + ATRi) cells in G1 per sample. Individual values and means ± SD are shown. P-values were determined by two-tailed unpaired t-test; *$p < 0.05$ (exact p-value is $p = 0.0209$), **$p < 0.01$ (exact p-value is $p = 0.0039$). **g** Quantification of the percentage of ALT-negative SW39 cells with RPA-marked lesions in G1 in different treatment conditions from $n = 3$ independent samples with $n_1 = 552$, $n_2 = 565$, $n_3 = 468$ (Control), $n_1 = 366$, $n_2 = 279$, $n_3 = 312$ (APH), $n_1 = 582$, $n_2 = 578$, $n_3 = 597$ (ATRi), $n_1 = 351$, $n_2 = 385$, $n_3 = 396$ (APH + ATRi) cells in G1 per sample. Individual values and means ± SD are shown. P-values were determined by two-tailed unpaired t-test; *$p < 0.05$ (exact p-value is $p = 0.0265$). Scale bar: 10 μm. A. U., arbitrary units. Source data are provided as a Source Data file.

G1 phase to prevent telomere damage and promote what we refer to as post-mitotic DNA synthesis (post-MiDAS). These findings extend the notion that DNA synthesis is not necessarily confined to the S-phase of the cell cycle[8]. Previous works had demonstrated that late replicating genomic regions continue DNA synthesis in G2/M[5,15,61,66]. Our results provide evidence that DNA replication continues even after cell division, implying that the fundamental biological processes of cell division and genome duplication can become temporally uncoupled (Fig. 7).

The ssDNA-binding protein RPA plays critical roles at replication forks during S-phase progression and to protect resected ssDNA upon chromosome breakage in S and G2. In contrast to the DNA damage response protein 53BP1, which is excluded from DNA breaks during mitosis[67,68], RPA can bind to exposed ssDNA in M-phase[69–71]. Together with our results, this suggests that RPA may shield replication stress-associated DNA lesions far beyond S-phase and exert its protective function at heritable lesions even in daughter cells.

RPA-marked lesions in newly born daughter cells are largely distinct from 53BP1 nuclear bodies marking replication stress-associated lesions such as CFSs. Consistent with recent work[72], our live cell experiments with endogenously tagged RPA show asymmetric distribution of RPA-marked lesions to daughter cells (Supplementary Movie 1), in contrast to symmetric distribution of 53BP1[17]. Moreover, CFS-associated 53BP1 nuclear bodies in G1 are induced upon inactivation of the DNA helicase BLM[17], whereas we find that RPA-marked telomere lesions in ALT cells are suppressed by BLM depletion. A fraction of 53BP1 nuclear bodies does co-localize with telomeres, however, and impaired RPA protection in G1 causes telomere damage and increases the fraction of 53BP1-positive telomeres (Fig. 6), suggesting that fragile chromosome ends can sporadically break and, as a second line of defense, recruit 53BP1 to prevent excessive generation of ssDNA by resection and avoid further attrition.

Why might fragile chromosome ends be processed differently from common fragile sites (CFSs) during cell division? On one hand, under-replicated regions far away from chromosome ends need well-coordinated enzymatic processing for sister chromatid

separation and completion of mitosis[6,70,73,74]. This may not be necessary, or at least not to the same extent, at more easily separable replication intermediates at the ends of chromosomes. Furthermore, highly repetitive regions such as telomeres may depend less on finishing replication before cell division, as opposed to non-repetitive, protein-coding sequences. Indeed, telomeres can use homology-directed telomere synthesis with non-sister telomeres as template[75]. In agreement, we find RPA-marked heritable telomere lesions associated with telomere maintenance primarily in ALT-positive cancer cells. ALT-positive cancer cells employ BIR and MiDAS in order to maintain their telomeres[49,60,61,66,76,77], and it will be important to determine in future work whether the post-mitotic DNA synthesis identified in our study uses a similar enzymatic machinery in G1 as the one employed by BIR in mitosis.

While previous studies made use of experimentally induced telomere-specific DNA breaks by fusing the FokI endonuclease to the telomere protein TRF1[66,75], cell cycle resolved high-content imaging allowed us to identify and investigate RPA-marked telomere lesions in ALT cells in unperturbed conditions, and thereby uncover telomere-specific post-mitotic DNA synthesis in G1. Identifying the cellular factors involved in post-MiDAS and elucidating their regulation will be an important task for future studies. Similarly, it will be interesting to investigate potential roles of the telomere-associated long non-coding RNA TERRA and of RNA-DNA hybrid homeostasis at telomeres in replication stress-associated formation of heritable RPA-marked lesions[52,78], in particular given that RNaseH1 depletion induces them (Supplementary Fig. 8). Moreover, the roles of RPA and ATR in TERRA R-loop regulation in different phases of the cell cycle deserve further attention.

Finally, as telomere maintenance by ALT represents a cancer-specific vulnerability, targeting post-MiDAS may offer new therapeutic opportunities. Specifically, ALT-positive cancer cells may use post-MiDAS as a backup mechanism to deal with unresolved telomere replication intermediates when telomere maintenance in S/G2 and during MiDAS at telomeres fails or is incomplete. Of note, ALT-positive cancers are not all alike, and

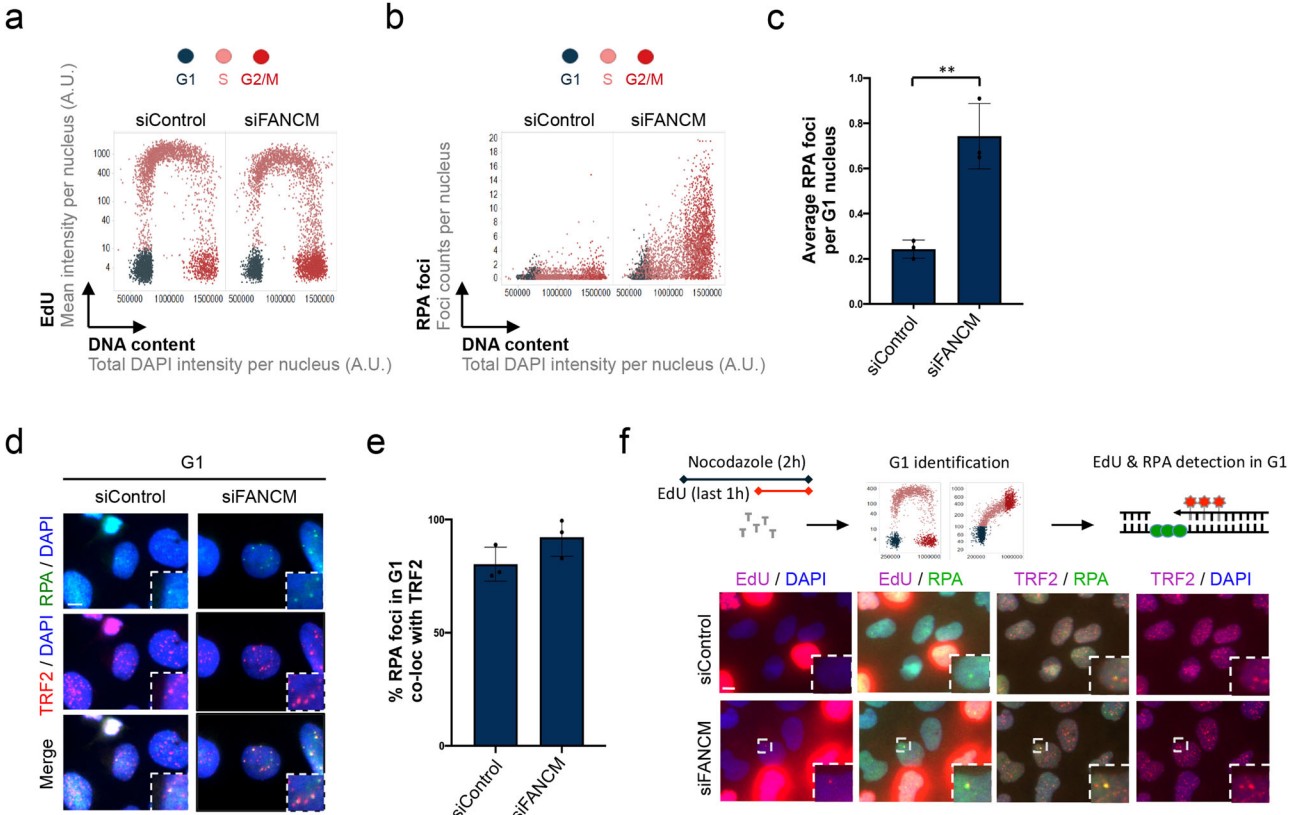

**Fig. 4 FANCM counteracts formation of RPA-marked inherited genomic lesions at telomeres. a** U-2 OS cells were transfected with siRNA for 48h as indicated. Cells were stained for EdU, RPA and TRF2. Cell cycle staging by EdU and DNA content is shown for at least 1000 cells per condition. **b** Cell cycle resolved RPA foci for the cells shown in **a**. **c** Quantification of average RPA foci counts per G1 nucleus in G1 in different treatment conditions from $n = 3$ independent samples with $n_1 = 1072$, $n_2 = 1065$, $n_3 = 1274$ (siControl), $n_1 = 898$, $n_2 = 916$, $n_3 = 951$ (siFANCM) cells in G1 per sample. Individual average values and means ± SD are shown. *P*-values were determined by two-tailed unpaired *t*-test; **$p < 0.01$ (exact *p*-value is $p = 0.0045$). **d** Representative images of G1 cells with RPA foci at telomeres. **e** Percentage of RPA foci co-localizing with TRF2 in G1 corresponding to **c**. Individual values and means ± SD are shown. **f** EdU incorporation at telomeres in G1 cells. G1 cells were identified in asynchronous U-2 OS cells by QIBC (2N DNA content and low EdU and/ or Cyclin A signal, cells marked in blue). EdU, RPA, and TRF2 signals were analyzed for DNA synthesis at telomeres in G1. EdU (20 μM, 1 h pulse) was added in the presence of nocodazole to exclude DNA synthesis events coming from S/G2 and from mitosis. Scale bars 10 μm. A. U., arbitrary units. Source data are provided as a Source Data file.

apart from differences in telomere maintenance may differ in replication timing and efficiency of cell cycle checkpoint activation, characteristics that in turn may impact lesion propagation through mitosis and the use of post-MiDAS. Telomerase-positive cancers can also have high loads of endogenous replication stress, and our results indicate that heritable RPA-marked DNA lesions occur in ALT-negative cancer cells in response to replication stress treatments. Thus, transgenerational telomere maintenance may not be limited to ALT, but similar mechanisms could be at play when cancer cells with reactivated telomerase experience replication stress. Thus, targeting post-MiDAS might have therapeutic benefits even beyond ALT-positive cancers.

## Methods

**Cell lines and drug treatments**. All cell lines were grown at 37 °C under standard cell culture conditions (humidified atmosphere, 5% $CO_2$). ALT-positive GM847, SW26, and U-2 OS cells, as well as derived cell lines, and ALT-negative HeLa, HeLa LT (long telomeres), SW39 and hTERT-RPE1 cells were grown in Dulbecco's modified Eagle's medium (DMEM) containing 10% fetal bovine serum (GIBCO) and penicillin-streptomycin antibiotics. For U-2 OS CycE/RAD52 WT and KO cells G418 400 μg/ml (Gibco, 10131-027), puromycin 1 μg/ml (Sigma, P8833) and tetracycline 2 μg/ml (Sigma, T7660) were added to the medium. U-2 OS 53BP1-GFP/RPA70-mScarlet cells were maintained in presence of puromycin 0.5 μg/ml (Sigma, P8833) and 5 μg/ml blasticidin (InvivoGen, ant-bl-05). All cells used in this study were grown under sterile conditions and routinely (monthly) tested for mycoplasma contamination and scored negative. The following compounds were used in this manuscript at the indicated final concentrations unless stated

otherwise: ATRi Az-20 (5 μM for RPE-1 cells, else 1 μM, Tocris), ATRi VE-821 (5 μM, Selleckchem), APH (0,2 μM, Sigma-Aldrich), HU (2 mM, Sigma-Aldrich), Nocodazole (50 ng/ml, Sigma-Aldrich), CDKi RO-3306 (9 μM, Sigma-Aldrich), Colcemid (0,1 μg/ml, Roche), ATMi KU-55933 (10 μM, Selleckchem).

**Cloning**. Cloning was done using chemically competent bacteria generated in-house, derived from Library Efficiency™ DH5α™ Competent Cells (ThermoFisher). All primers used for cloning are provided in Supplementary Table 1. Correct cloning and integration into target vectors were confirmed by sequencing.

**Cloning of components for endogenous RPA70 tagging**. The pUC18 RPA70 ha800 mScarlet-P2A-Blast construct was generated in two steps. First, pUC18 RPA70 ha800 was generated by two-piece Gibson assembly consisting of pUC18 linearized with primers 1 and 2 and the *RPA70* homology fragment spanning around 800 bp upstream and downstream of the *RPA70* stop codon, amplified with primers 3 and 4. In the second step, using Gibson assembly, the linearized destination vector pUC18 RPA70 ha800 (primers 5 and 6) was combined with mScarlet-P2A-Blast fragment amplified from pUC18 mScarlet-P2A-BlastR vector (primers 7 and 8). RPA70 gRNA was cloned into SpCas9 expressing vector pX459, targeting the C-terminus of RPA70 as previously described[79]. Primers 9 and 10 were phosphorylated and annealed at 10 μM using T4 PNK. The product was diluted and further assembled into the pX459 vector by golden gate assembly using BbsI and T4 DNA ligase (12 cycles of 5 min at 37 °C and 5 min at 16 °C) followed by transformation and isolation of plasmids to identify the correct product.

**Engineering of the endogenous *RPA70* gene locus**. 350'000 U-2 OS GFP-53BP1 cells were seeded into individual wells of a 6-well plate (desired confluence at the time of transfection 80–90%). After 24 h, 1 μg of the pX459 targeting RPA70 and 1 μg of the repair template (pUC18 RPA70 ha800 mScarlet-P2A-Blast plasmid

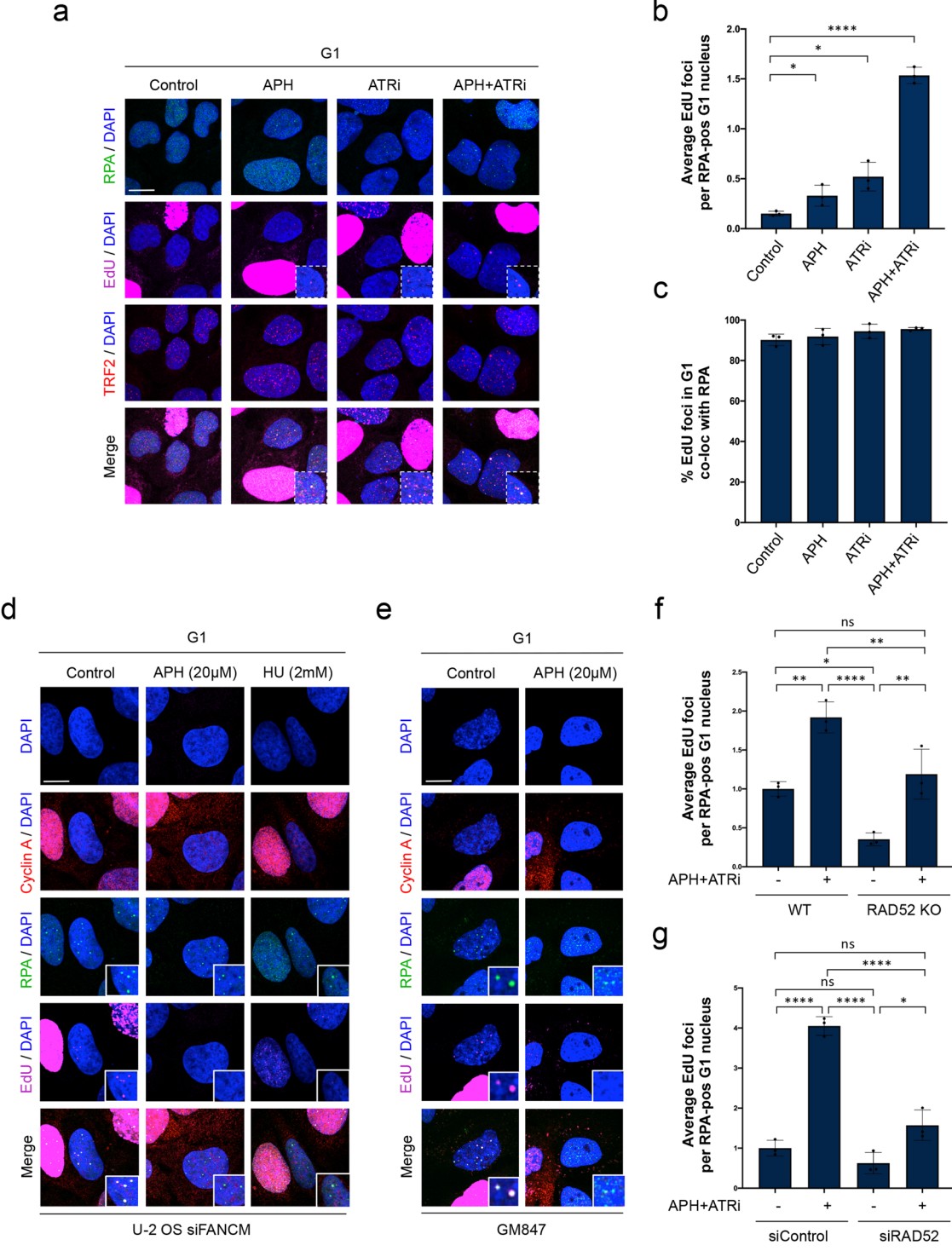

containing the repair template), were diluted in 250 µl OptiMEM, followed by addition of 6 µl TransIT-LT1 and 15 min incubation. The transfection mix without repair template was used as control. One day after transfection cells were transferred to 15 cm plates. 48 h upon transfection selection was initiated by incubation with 5 µg/ml blasticidin. The medium was exchanged every 3 days until colonies were obtained. Individual clones were picked by trypsin detachment and transferred to a 24-well plate. These were further expanded under selection to 6-well plates and characterized by PCR-based analysis of genomic DNA to confirm the correct insertion of the mScarlet-P2A-Blast module (primers 11 and 12) and by QIBC for fusion protein expression and functionality.

**siRNA transfections**. Individual siRNA transfections were performed with Ambion Silencer Select siRNAs using Lipofectamine RNAiMAX (Thermo Fisher Scientific). The following Silencer Select siRNAs were used at a concentration of

25 nM: siRPA70 (s12127), siRPA32 (s12130), siFANCM (s33621), siBLM (s1999), siRNASH1 (s48358), siASF1A (s24604), siASF1B (s31345). For RAD52 depletion Qiagen siRAD52 (SI02629865) was used. For RPA depletion in G1 cells, a 50 nM siRNA concentration was used. Silencer Select control (s813) from Ambion was used as non-targeting control.

**Immunostaining**. Cells were grown on sterile 12 mm glass coverslips, fixed in 3% formaldehyde in PBS for 15 min at room temperature, washed once in PBS, permeabilized for 5 min at room temperature in 0.2% Triton X-100 (Sigma-Aldrich) in PBS, and washed twice in PBS. All primary and secondary antibodies were diluted in filtered DMEM containing 10% FBS and 0.02% Sodium Azide. Antibody incubations were performed for 1–2 h at room temperature. Following antibody incubations, coverslips were washed once with PBS and incubated for 10 min with PBS containing 4′,6-Diamidino-2-Phenylindole Dihydrochloride (DAPI, 0.5 µg/ml)

**Fig. 5 RPA-marked lesions are primed for post-mitotic DNA synthesis (post-MiDAS). a** Asynchronously growing U-2 OS cells were treated for 24h as indicated and EdU foci (100μM, 1h pulse) in G1 cells were analyzed together with TRF2. Representative confocal images are shown. **b** Quantification of average EdU foci counts per RPA-positive G1 nucleus in different treatment conditions from $n = 3$ independent samples with $n_1 = 264$, $n_2 = 260$, $n_3 = 275$ (Control), $n_1 = 94$, $n_2 = 76$, $n_3 = 86$ (APH), $n_1 = 343$, $n_2 = 379$, $n_3 = 330$ (ATRi), $n_1 = 291$, $n_2 = 312$, $n_3 = 350$ (APH + ATRi) cells in G1 per sample. Individual average values and means ± SD are shown. $P$-values were determined by two-tailed unpaired t-test; *$p < 0.05$ (exact p-values are $p = 0.0445$ and $p = 0.0439$, respectively), ****$p \leq 0.0001$. **c** Percentage of EdU foci co-localizing with RPA in G1 corresponding to **b**. Individual values and means ± SD are shown. **d** U-2 OS cells were treated as in Fig. 4f, without or with APH (20 μM, 2h) or HU (2 mM, 2h) to block DNA synthesis. EdU (20 μM, 1h pulse) and RPA signals were analyzed in Cyclin A-negative G1 cells by confocal microscopy. **e** EdU (20 μM, 1h pulse) and RPA signals in G1 in untreated Cyclin A-negative GM847 cells analyzed by confocal microscopy. APH (20 μM, 2h) as in **d** was added as indicated to block DNA synthesis. **f** Quantification of average EdU foci counts (100 μM, 1h pulse) per G1 nucleus in wild-type (WT) and RAD52 knockout (KO) U-2 OS cells, treated or not for 24 h with APH (0.2 μM), ATRi (1 μM) from $n = 3$ independent samples with $n_1 = 312$, $n_2 = 415$, $n_3 = 375$ (WT), $n_1 = 419$, $n_2 = 449$, $n_3 = 348$ (WT + APH + ATRi), $n_1 = 135$, $n_2 = 242$, $n_3 = 167$ (KO), $n_1 = 240$, $n_2 = 369$, $n_3 = 380$ (KO + APH + ATRi) cells in G1 per sample. Data were normalized to untreated control and individual average values and means ± SD are shown. $P$-values were determined by one-way analysis of variance (ANOVA) with Tukey's test; *$p < 0.05$ (exact $p$-value is $p = 0.0167$), **$p < 0.01$ (exact $p$-values are $p = 0.0021$, $p = 0.0087$, and $p = 0.0038$, respectively), ****$p \leq 0.0001$, ns $p \geq 0.05$ (exact $p$-value is $p = 0.6632$). **g** Quantification of average EdU foci counts (100 μM, 1h pulse) per G1 nucleus in U-2 OS cells transfected with siControl or siRAD52, treated or not for the last 24 h with APH (0.2 μM) and ATRi (1 μM) from $n = 3$ independent samples with $n_1 = 181$, $n_2 = 321$, $n_3 = 289$ (siControl), $n_1 = 443$, $n_2 = 471$, $n_3 = 444$ (siControl + APH + ATRi), $n_1 = 356$, $n_2 = 394$, $n_3 = 394$ (siRAD52), $n_1 = 445$, $n_2 = 445$, $n_3 = 431$ (siRAD52 + APH + ATRi) cells in G1 per sample. Data were normalized to untreated control and individual average values and means ± SD are shown. $P$-values were determined by one-way analysis of variance (ANOVA) with Tukey's test; *$p < 0.05$ (exact value is $p = 0.0133$), ****$p \leq 0.0001$, ns $p \geq 0.05$ (exact $p$-values are $p = 0.4064$ and $p = 0.1312$, respectively). Scale bars 10 μm. A. U., arbitrary units. Source data are provided as a Source Data file.

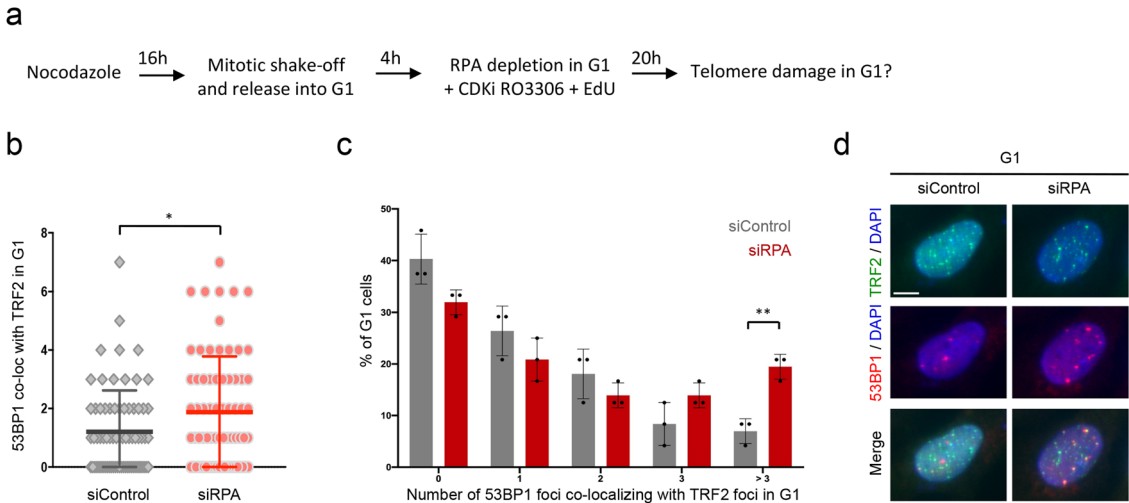

**Fig. 6 RPA protects ALT telomeres during post-MiDAS from breakage. a** General scheme of the experimental design to deplete RPA in G1 cells. Briefly, U-2 OS cells were arrested in G2/M by nocodazole, a mitotic shake-off was performed and cells were released in fresh medium for 4 h, allowing them to pass through mitosis. RPA depletion was then performed for 20 h in presence of CDKi RO-3306 (to block remaining S/G2 cells from cell division) and EdU (to mark cells entering S-phase and to be able to discriminate them from G1 cells), in order to assess telomere damage in G1. **b** Telomeric DNA damage in G1 as measured by 53BP1 co-localization with TRF2 in $n = 72$ G1 cells. Horizontal lines indicate means ± SD of the depicted single cell data; two-sided Mann–Whitney $U$ test was performed. *$p < 0.05$ (exact $p$-value is $p = 0.0469$). **c** Percentage of G1 cells with damaged telomeres upon RPA depletion from $n = 3$ independent samples with $n = 24$ cells per sample. Control cells are in gray, RPA-depleted cells in red. Individual values and means ± SD are shown. $P$-values were determined by two-tailed unpaired $t$-test **$p < 0.01$ (exact $p$-value is $p = 0.0031$). **d** Representative images of G1 cells with telomeric DNA damage upon RPA depletion in G1. Scale bar 10 μm. A. U., arbitrary units. Source data are provided as a Source Data file.

at room temperature to stain DNA. Cells were subsequently washed three times in PBS and briefly submerged in distilled water prior to being mounted on glass slides with 5 μl Mowiol-based mounting media (Mowiol 4.88 in Glycerol/TRIS). The following primary antibodies were used in this study: RPA70 (rabbit, Abcam ab79398, 1:500), RPA32 (mouse, Abcam ab2175, 1:500), TRF2 (rabbit, Biolegend NB110-57130, 1:500), TRF2 (mouse, Abcam ab13579, 1:500), 53BP1 (mouse, Upstate MAB3802, 1:500), 53BP1 (rabbit, NB100-304 Novus, 1:1000), Cyclin A (rabbit, Santa Cruz sc-751, 1:100), Cyclin A (mouse, Abcam ab16726, 1:200), PML PG-M3 (mouse, Santa Cruz sc-966, 1:100), H2AX Phospho S139 (mouse, Biolo-gend 613401, 1:500). To stain RAD52[80], cells were fixed in ice-cold methanol for 15 min at −20 °C. Cells were permeabilized in PBS containing 0.2% Triton X-100 (Sigma-Aldrich). The permeabilized cells were blocked with PBS containing 1% BSA and then processed as described above. Secondary antibodies and dilutions were: Alexa Fluor 488 Goat Anti-Rabbit (Life Technologies, A11034, 1:500), Alexa Fluor 488 Goat Anti-Mouse (Life Technologies, A11029, 1:500), Alexa Fluor 568 Goat Anti-Rabbit (Life Technologies, A11036, 1:500), Alexa Fluor 568 Goat Anti-Mouse (Life Technologies, A11031, 1:500), Alexa Fluor 647 Goat Anti-Rabbit (Life Technologies, A21244, 1:500), Alexa Fluor 647 Goat Anti-Mouse (Life

Technologies, A21235, 1:500), Donkey anti-Sheep IgG (H L) Cross-Adsorbed Secondary AB Alexa 488 (Life Technologies, A11015, 1:500).

**DNA-FISH**. After immunofluorescence or metaphase spreads preparation, for DNA-FISH samples were dehydrated by incubation with 70%, 95 and 100% ethanol, each 5 min at RT. Samples were air-dried. Onto each coverslip 20 μl of hybridization mix (10 mM Tris-HCL pH 7.4, 70% formamide, 0,5% blocking reagent Roche 11096176001, 1:1000 of TelC-Cy3 probe F1002 or 1:2000 of TelG-Cy5 probe F1007, PNA Bio) was added. Samples were then denatured at 80 °C for 10 min (the denaturation step was omitted when assessing ssDNA at telomeres) and hybridized for 3 h at RT in the dark. Samples were further washed twice with wash buffer A (10 mM Tris-HCL pH 7.4, 70% formamide) for 15 min and 3 times with wash buffer B (0.1 M Tris-HCL pH 7.4, 0.15 M NaCl, 0.08% Tween-20), 5 min each. Following these washing steps, samples were incubated for 10 min with PBS containing DAPI (0.5 μg/ml), washed three times in PBS, dehydrated by incubation with 70%, 95 and 100% ethanol 5 min each, and mounted.

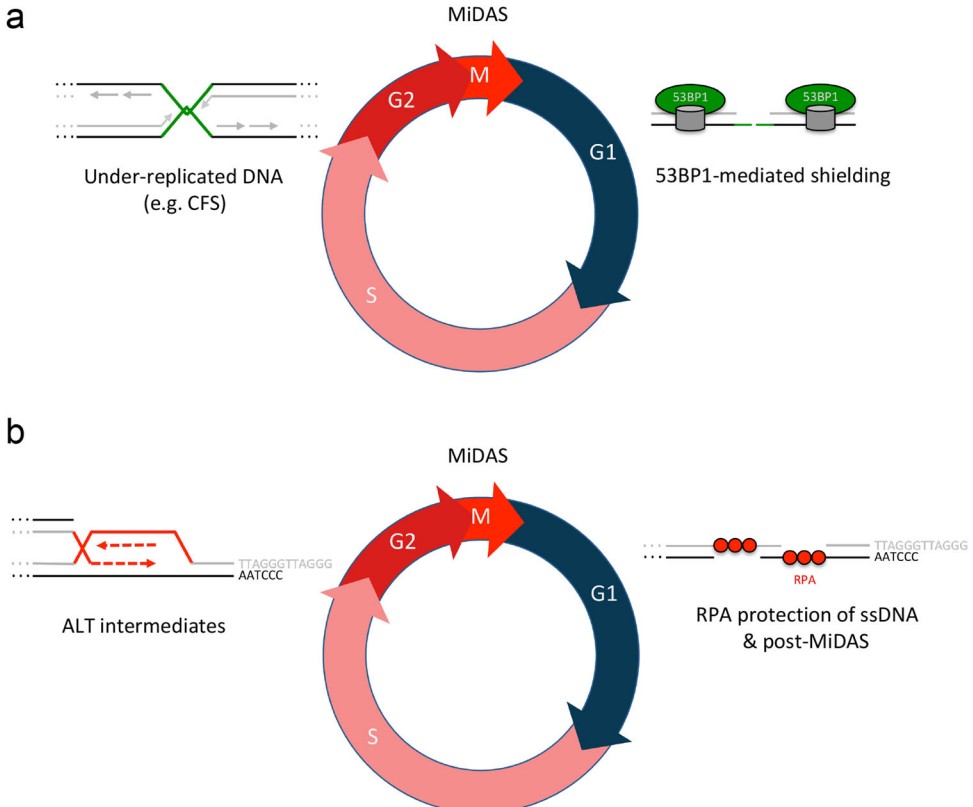

**Fig. 7 Model of inherited ssDNA lesions at telomeres, shielded by RPA for post-MiDAS. a** Under-replicated DNA, e.g., at common fragile sites (CFS), is transmitted from S/G2 to mitosis, giving rise to 53BP1 nuclear bodies in G1. **b** ssDNA lesions at chromosome ends, e.g., ALT intermediates, are transmitted from S/G2 to mitosis and to G1, where they are protected by RPA to allow for post-mitotic DNA synthesis (post-MiDAS). Replication stress in the previous cell cycle and/or incomplete MiDAS exacerbate heritable telomere lesions, and failure to maintain telomere integrity in the subsequent G1 phase, e.g., by loss of RPA protection, leads to telomere damage and elevated association of 53BP1 with broken telomeres.

**EdU labeling and CldU incorporation**. For pulsed EdU (5-ethynyl-2′-desoxyur-idine) (Thermo Fisher Scientific) labeling, cells were incubated for 20 min in medium containing 10 µM EdU, unless stated differently in the figure legends. For EdU foci detection in G1, replication stress treatments were washed out and the EdU pulse was performed in fresh medium. The Click-iT EdU Alexa Fluor Imaging Kit (Thermo Fisher Scientific) was used for EdU detection. For detection of single-stranded DNA with CldU, cells were incubated with 40 µM CldU (c6891, Sigma-Aldrich) for 48 h (last 24 h in presence of replication stress treatments). Cells were pre-extracted for 5 min on ice using CSK buffer (10 mM Pipes pH 7.0, 100 mM NaCl, 300 mM sucrose, 3 mM MgCl$_2$, 0.5% TX-100), fixed in 3% formaldehyde in PBS for 15 min at room temperature, washed once in PBS, permeabilized for 10 min at room temperature in PBS supplemented with 0.2% Triton X-100 (Sigma-Aldrich), followed by 20 min blocking at RT in filtered 3% BSA in PBS. The following steps were then executed according to the standard immunostaining protocol described above. Primary antibody used was rat anti-BrdU (recognizing CldU, Abcam ab6326, 1:500), secondary antibody was Cy3 donkey anti-rat1 (712-166-153, Jackson Immuno-Research Laboratories, Inc, 1:500).

**Sequential re-staining**. For sequential re-staining and iterative indirect immu-nofluorescence imaging (4i)[45] cells were seeded in 96-well plates (Greiner CELL-STAR 96-well-plates, Sigma-Aldrich) and after desired treatments fixed with 4% PFA for 15 min, washed with 1 x PBS, permeabilized for 15 min with 0,5% Triton X-100 in PBS. Blocking was done with sBS (1% Bovine Serum Albumine (BSA) in PBS with 150 mM Maleimide). Primary and secondary antibodies were diluted in cBS (sBS without maleimide) and applied for 1–2 h. Staining with DAPI (0.5 µg/ml) was applied as described above. The imaging acquisition was performed in imaging buffer (700mM N-Acetylcysteine (NAC) in ddH$_2$O, pH 7.4). After the first acquisition, for every subsequent staining these steps were applied: 5 washes with H$_2$O, followed by antibody elution in elution buffer EB (0.5M L-Glycine pH 2.5, 3 M Urea, 3 M Guanidinum chloride (GC), and 70 mM TCEP HCl (TCEP) in ddH$_2$O) on a shaker for 10 min. EB was washed out three times with sBS and the samples were blocked for 1 h. Antibody application, DAPI staining and acquisition were performed as for the initial staining.

**Quantitative image-based cytometry (QIBC)**. Automated multichannel wide-field microscopy for quantitative image-based cytometry (QIBC)[81,82] was performed on the Olympus ScanR Screening System equipped with an inverted motorized Olympus IX83 microscope, a motorized stage, IR-laser hardware autofocus, a fast emission filter wheel with one set of bandpass filters for multi-wavelength acquisition: DAPI (ex BP 395/25, em BP 435/26), FITC (ex BP 470/24, em BP 511/23), TRITC (ex BP 550/15, BP 595/40) and Cy5 (ex BP 640/30, em BP 705/72), and a Hamamatsu ORCA-FLASH 4.0 V2 sCMOS camera (2048 × 2048 pixel, pixel size 6.5 µm x 6.5 µm). The associated objectives used were a 20x NA 0.75 UPLSAPO and a 40x NA 0.90 UPLSAPO air objective. For each condition, image information of large cohorts of cells (typically at least 500 cells for the UPLSAPO 40x objective (NA 0.9), at least 1000 cells for the UPLSAPO 20x objective (NA 0.75)) was acquired under non-saturating conditions. Identical settings were applied to all samples within one experiment. Images were analyzed with the Olympus ScanR Image Analysis Software (versions 3.0.0 & 3.0.1), a dynamic background correction was applied, and nuclei segmentation was per-formed using an integrated intensity-based object detection module using the DAPI signal. Foci segmentation was performed using an integrated spot-detection module. All downstream analyses were focused on properly detected interphase nuclei containing a 2C-4C DNA content as measured by total and mean DAPI intensities. Fluorescence intensities were quantified and are depicted as arbitrary units. Color-coded scatter plots of asynchronous cell populations were generated with Spotfire data visualization software version 7.0.1 and 10.10.1 (TIBCO). Within one experiment, similar cell numbers were compared for the different conditions. For visualizing discrete data in scatter plots, mild jittering (random displacement of data points along the discrete data axes) was applied in order to demerge over-lapping data points. Representative scatter plots, typically containing several hundred to several thousand cells each, are shown. Representative images, in which individual color channels have been adjusted for brightness and contrast, accom-pany selected quantifications. G1 cells were identified based on the total DAPI and the mean nuclear EdU and/or Cyclin A intensities. Where the experimental setup did not allow for a combination of cell cycle markers, cells were gated based on total DAPI intensities and the results were validated in separate control experi-ments with EdU and/or Cyclin A. Manual quantification of foci co-localization in G1 cells was performed on G1-gated or RPA foci positive (i.e. ≥ 1 detected RPA focus) G1 subpopulations on the Olympus ScanR Image Analysis Software. For automated co-localization analysis, the exported foci parameters (x/y center posi-tions and area) were used to identify the number of co-localizations per cell in Spyder 3.3.6. Each parameter file was turned into a dictionary, with parent object

ID as the key and x/y coordinates and foci areas as values. Each foci type A was then compared with all the foci type B per nucleus. The distance between two foci was calculated using the x/y center positions according to Pythagoras. If the distance between two foci centers was smaller than the radius of either of them (assuming spherical foci shapes), this was considered co-localization.

**Time-lapse microscopy**. Time-lapse microscopy was performed either on a GE IN Cell Analyzer 2500HS system (Supplementary Fig. 3c) or on a Molecular Devices ImageXpress IXM-C (Supplementary Fig. 3d) under $CO_2$ (5%) and temperature (37 °C) control. The IN Cell Analyzer 2500HS is equipped with two filter sets for multi-wavelength acquisitions: BGOFR_1, blue (ex BP 390/18, em BP 432/47), green (ex BP 475/28, em BP 511/23), orange (ex BP 542/27, em BP 587/48) and farred (ex BP 632/22, em BP 676/48), and BGFFR_2, blue (ex BP 390/18, em BP 432/47), green (ex BP 475/28, em BP 526/52), red (ex BP 575/25, em BP 607/19) and farred (ex BP 632/22, em BP 676/48). The objective used was a 20x NA 0.75 CFI Plan Apo lambda. Images were acquired with a 16-bit 2048 × 2048 pixel PCO sCMOS camera with a pixel size of 6.5 μm. The IXM-C system is equipped with the following bandpass filters for multi-wavelength acquisition: DAPI (ex BP 377/25, em BP 447/30), FITC (ex BP 480/15, em BP 535/20), TRITC (ex BP 542.5/7.5, em BP 642.5/32.5), Texas Red (ex BP 560/20, em BP 630/30) and Cy5 (ex BP 624/20, em BP 692/20), and with lasers at 405, 488, 532, and 633 nm. The objective used was 20x NA 0.75 Apo lambda. Images were acquired with a 16-bit 2048 × 2048 pixel Andor sCMOS camera with pixel size of 6.5 μm. For time-lapse microscopy, cells were plated on 96-well ibidi plates at a density of 5000–20000 cells per well 24 h prior to imaging in FluoroBrite DMEM medium containing 10% FCS (GIBCO) and penicillin-streptomycin. Images were taken at 30 min intervals for up to 72 h. Image stacks were generated and processed with Fiji (ImageJ 64-bit; Version 2.00-rc-54/1.51 h).

**Confocal microscopy**. Confocal images were acquired on an automated Leica SP5 inverted confocal laser scanning microscope equipped with Argon lasers for 453 nm, 476 nm, 488 nm, 496 nm and 514 nm, and a diode laser for 561 nm. Confocal images were acquired with an HCX PL APO Leica 63x oil immersion objective with PMT detectors using Leica Application Suite X 3.6.0.20104 and Leica LAS AF Version 2.7.3.9723.

**RPA depletion in G1**. Asynchronously growing U-2 OS cells were synchronized in G2/M in presence of nocodazole (50 ng/ml) for 16 h. Mitotic cells were collected by shake-off, washed twice in PBS and re-seeded on coverslips. After cells were allowed to progress into the subsequent G1 phase for 4 h, RPA depletion was performed for 20 h in presence of CDKi RO3306.

**Metaphase spreads**. U-2 OS cells were treated with replication stress-inducing drugs, released into fresh medium for 20 h after which colcemid was added at a concentration of 0.1 μg/ml for 4 h. The medium was collected and cells were harvested by trypsinization, samples were collected by centrifugation, and the medium was removed leaving approximately 1 ml. Mitotic cells were resuspended at a concentration of $10^6$ cells per ml in pre-warmed hypotonic solution (10 mM Tris pH7.5, 10 mM NaCl, 5 mM MgCl₂). Cell suspensions were further diluted 1:4 with hypotonic solution and the cells were spread on slides using Cytospin. 250–500 μl of cell suspension was loaded per spot. The slides were spun at 113xg for 5 min. The slides were left to dry at room temperature and then immediately immersed into ice-cold fixative solution (4% paraformaldehyde, 1× PBS, 0.1% Triton X-100) at 4 °C for 20 min. Then they were washed with ice-cold 1× PBS for 3 times. The cells were permeabilized with 0.2% Triton X-100 in 1× PBS at 4 °C for 15 min. The slides were then washed again with ice-cold 1× PBS for 3 times.

**Southern blot**. Genomic DNA was extracted and 1 μg of HinfI/RsaI-digested DNA was run on a 0.6% agarose gel. The gel was transferred to a positively charged nylon membrane and probed with a $^{32}$P-labeled telomere-probe[83].

**RNA extraction, reverse transcription, and quantitative PCR**. RNA was purified with TRIzol reagent (Life Technologies), primed with random hexamers (11034731001, Roche) and reverse-transcribed using MultiScribe Reverse Transcriptase (4311235, Thermo Fisher). Quantitative PCR (qPCR) experiments to control knockdown efficiencies were performed in technical triplicates with the KAPA SYBR FAST qPCR Kit (KAPA Biosystems) on a Rotor-Gene Q system (Qiagen). Relative transcription levels were obtained by normalization to RPS12. Primers used for amplification are provided in Supplementary Table 2.

**Statistics and reproducibility**. Graph Pad Prism version 8 was used for statistical analysis. Statistical analysis was performed by two-tailed unpaired t-test, two-sided Mann-Whitney-U test, or analysis of variance (ANOVA) with Tukey's test. Sample sizes and the statistical tests used are specified in the figure legends. Micrographs in Figs. 1d, 2f, 3a, 4d, f, 5a, d, e, 6d and Supplementary Figs. 1a, 2c, 3c, d, 4c, 5c, 6b, c, 7a, 8o–q, 9f, g, 10a show representative examples from experiments that were repeated independently at least 3 times with similar results. Micrographs in Supplementary Fig. 7e show representative examples from experiments that were

repeated independently at least twice with similar results. The control experiments shown in Supplementary Figs. 3a, 6a, 8a, b were performed once. The QIBC data in Figs. 1a, b, c, 2b–e, 3a–c, 3e, 4a–c, e, 5b, c, 6b, c, d and Supplementary Figs. 1a–d, g–l, j, 4a–d, 5a, b, e, f, 7b–e, f, 8c, d, e, g, l, 9a, e, 10a, f, g are from experiments that were repeated independently at least 3 times with similar results. The QIBC data in Figs. 3d, f, g, 5f, g and Supplementary Figs. 1e, f, 2a, b, c, 3b, 7d, e, 8f, j, n, 10b, d, e are from experiments that were repeated independently at least twice with similar results.

**Reporting summary**. Further information on research design is available in the Nature Research Reporting Summary linked to this article.

## Data availability
Numerical source data underlying Fig. 1a–c; 2b–e; 3a–g; 4a–c, e; 5b, c, f, g; 6b, c and Supplementary Figs. 1a–h; 2a–c; 3b; 4a–d; 5a, b, e, f; 7b–e; 8a–n; 9a–e; 10a–g, and an uncropped scan of the Southern blot shown in Supplementary Fig. 8b are provided in the Source Data file linked to this paper. Primary imaging data have been deposited at the European Bioinformatics Institute (EBI) BioStudies database (https://www.ebi.ac.uk/biostudies/) with accession number S-BIAD106. There are no restrictions on data availability. Source data are provided with this paper.

## Code availability
Source code for foci co-localization is deposited at GitHub and can be accessed at https://github.com/AltmeyerLab/Nature-Comm_Lezaja-et-al._2021.

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

## Acknowledgements

We acknowledge the University of Zurich Center for Microscopy and Image Analysis for technical support. We would like to thank Sinan Kilic for help with cloning and genome editing, Michaela Remisova for help with co-localization analyses, and Aleksandra Vancevska for advice on telomere FISH experiments. Claus Azzalin and Antonio Porro kindly provided GM847 cells, Jiri Lukas kindly provided U-2 OS 53BP1-GFP cells, and Luis Toledo kindly provided U-2 OS RPA-GFP cells. Jerry Shay kindly provided SW26 and SW39 cells, and Thanos Halazonetis kindly provided inducible U-2 OS CycE/RAD52 WT and KO cells and RAD52 antibody. This work was supported by the Swiss National Science Foundation (PP00P3_150690 & PP00P3_179057), the European Research Council (ERC) under the European Union's Horizon 2020 research and innovation program (ERC-2016-STG 714326), the Novartis Foundation for Medical-Biological Research (16B078), and the Swiss Foundation to Combat Cancer (Stiftung zur Krebsbekämpfung).

## Author contributions

A.L. initiated the project, designed and conducted most of the experiments, analyzed, interpreted and visualized results, and developed the study. A.P. and Y.W. contributed to QIBC and qPCR experiments and data analysis, E.C. performed Southern blots, R.I. generated reagents and contributed to cloning. M.A. conceived and supervised the study. A.L. and M.A. wrote the manuscript with input and edits from all authors.

## Competing interests

The authors declare no competing interests.
