## [Peer Review File · Nature Communications]

REVIEWER COMMENTS

Reviewer #1 (Remarks to the Author):

In the manuscript entitled „RPA shields inherited DNA lesions and primes fragile telomeres for post-mitotic DNA synthesis“, Lezaja et al, provide evidence that RPA foci which can be visualized in the G1 phase of the cell cycle are largely due to single stranded gaps at telomeres. They go on to demonstrate that these gaps are due to replication problems that have arisen in the previous cell cycle. Importantly, the RPA foci are distinct from 53BP1 foci. The RPA foci co-localize with ssDNA. Interestingly, in the 4 cell lines that were investigated, it appears that only cells employing ALT are prone to increased telomeric RPA foci in G1. The G1 ssDNA RPA foci co-localize with PML bodies, where it is presumed that ALT telomeres get elongated. By increasing replication stress, either through aphidicolin treatment or the depletion of FANCM, the number of G1 RPA foci could be increased. Finally, the authors demonstrate that the presence of RPA at telomeres is important to prevent the association of 53BP1. In a model the authors present a model that suggests that recombination intermediates in ALT cells get passed onto G1 where they serve to protect ssDNA and promote priming for extension. In general, I find that this is an important contribution to the field and I recommend that it is published in Nat. Comm. I do have a few concerns and suggestions that I think could be addressed before publication occurs. I feel that all experiments have been carefully controlled and that the appropriate statistical tests were applied.

Suggestions:

1. Since only 4 cell lines were employed, I find it difficult to conclude that this only occurs in ALT cell lines. The authors could try expressing telomerase in the ALT cells and see if the RPA foci diminishes. Alternatively, shAsf1 in HeLa cells with long telomeres could be employed to induce isogenic ALT cells (see O'sullivan et al, NSMB, 2014)
2. In the experimental set up shown in figure 5, are there more end-to-end fusions seen in metaphase spreads, this would be expected, but it would be nice to see.
3. In the discussion the authors speculate about the source of replication stress in ALT cells stemming from TERRA. This seems reasonable given the phenotypes seen with siFANCM, which reduced R-loops at telomeres. Does the overexpression of RNase H1 reduce RPA foci in G1, this would be easy to test and would hint towards the source of the G1 foci.
4. Building on point 3. Does the depletion of RNase H1 increase the RPA foci.
5. Finally, if the recombination intermediates are the source of the RPA foci then depletion of RAD52 should also reduce them.

Reviewer #2 (Remarks to the Author):

In this manuscript, the authors describe a role for RPA in protecting telomeric DNA lesions that arise from replication stress, to facilitate post-mitotic DNA repair. Given the structural complexities of telomeric DNA, telomeres are prone to replication stress. Moreover, replication stress is exacerbated in cells that rely on the ALT pathway for telomere maintenance. ALT telomeres have been demonstrated to rely on break-induced replication (BIR) to not only mitigate replication stress, but to also promote telomere elongation. Mechanistically, BIR has been demonstrated to occur both in interphase, where it is referred to as break induced telomere synthesis (BiTS), as well as metaphase, where it is referred to as mitotic DNA synthesis (MiDAS). Here, the authors describe another type of DNA repair mechanism that they refer to as 'post-MiDAS', which represents RPA-coated ssDNA lesions that are repaired in G1. The authors present data to suggest that the type of repair that arises at these DNA lesions is dictated in part by the binding of either 53BP1 or RPA proteins. By exacerbating replication stress at ALT telomeres the authors demonstrate that both proteins bind DNA lesions in G1 independently of one another, but that there is a substantial increase in RPA foci at telomeric DNA suggesting an increase in telomeric

ssDNA gaps. The authors conclude that RPA and 53BP1 may dictate the type of repair that occurs at these heritable lesions with ALT cells being more reliant on RPA.

Fully defining the events that lead to repair of DNA lesions at telomeres is critically important to the understanding of telomere maintenance in ALT and would be of broad interest to the scientific community. However, the data presented in the manuscript almost entirely lack statistical rigor and do not provide enough evidence to support the conclusions of the manuscript.

Major Points

1. The terminology the authors use to describe the repair mechanism at ALT telomeres is confusing. MiDAS refers to DNA synthesis events that arise during mitosis. Therefore, post-MiDAS would imply an event that happens after mitotic DNA synthesis. This leads to the confusion. Are the authors suggesting that ssDNA gaps that arise as a byproduct of MiDAS are then further repaired in G1 by BIR? Or are the authors suggesting that RPA coated ssDNA gaps generated during S-phase traverse mitosis and are then repaired in G1. The latter is more consistent with the tone of the manuscript suggesting that the reference should be to post-mitotic repair, not post-MiDAS.
2. The entire manuscript is based on the use of U2OS and GM847 cell lines. These cell lines both have inherently low levels of 53BP1 foci (less than 10% of cells have greater than 10 foci), the authors should use an ALT cell line with higher 53BP1 foci to see if this general trend is consistent or unique to this subset of cells. This discrepancy in 53BP1 foci is noted in the literature, but the 'why' is still unclear. SAOS2 and SKLU1 are two ALT cell lines that are commercially available and approximately 70% of these cells have greater than 10 53BP1 foci per cell suggesting that these cells may in fact respond differently to DNA damage. The authors should use these cell lines in addition to the U2OS and GM847 to see whether the phenotypes are similar. It is possible, maybe even likely, that the results between these groups of ALT cells may be different suggesting that ALT cells come in different 'flavors'. Analyzing the SW26 (ALT+), SW39 (ALT-), and IMR90 (parental) isogenic cell lines in this assay would also better support this analysis.
3. The lack of statistical rigor throughout the manuscript is surprising. All of the data should be analyzed using the proper methods for statistical analysis. In addition, the analysis of such few cells (20 cells in some experiment) is insufficient.
4. The data in Figure 1b suggest that upon replication stress the number of RPA foci per nucleus ranges from 0 foci to 20. The graph in Figure 1c averages those foci and determines that on average a G1 nucleus will have 1 foci per cell +/- SD of 0.5. This analysis is not consistent with the data presented by QIBC in Figure 1b, Extended Data Figure 1d, and Figure 2b. In fact, the number of RPA foci vary widely between these figures, this discrepancy is concerning.
5. The RPA70 foci formation in G1 (Figure 1B) appear to be far more robust than the RPA32 foci formation (Extended Data Figure 1E) in G1 suggesting a discrepancy in the data. Analyzing both RPA subunits side-by-side in QIBC with statistical analysis would help to clarify whether this is actually significant. The authors need to clarify.
6. If replication stress exacerbates ssDNA formation at telomeres and this is marked by RPA in G1, and RPA foci increase in G1, why don't the percentage of RPA/TRF2 colocalization events increase in Figure 2d and Extended Figure 5D. This argues that the vast majority of RPA foci that accumulate following replication stress, do not arise at telomeric DNA. This directly contradicts the main conclusions from the manuscript and should be clarified.
7. If telomeres have an increase in ssDNA at telomeres following ATRi and APH treatment the authors should analyze this by S1 nuclease treatment combined with TRF analysis.
8. The conclusion that RPA foci represent sites of ssDNA gaps is not supported by experimental data. These RPA foci could represent many different types of DNA lesions that contain single-stranded DNA, to specifically refer to them as ssDNA gaps would require additional experimental

evidence. The data demonstrating colocalization between TelC and RPA, and TelG and RPA, could suggest that the probe is binding one strand and RPA is binding the other, which would indicate a bubble structure, not simply ssDNA. Supporting this, the TelG and TelC show strong correlation under native conditions suggesting both strands are present, but in the form of ssDNA. There are too many possibilities to reliably state that the lesions are specifically ssDNA gaps.

9. The authors present a model in Figure 6 to suggest that 53BP1 protects fragile sites in G2/M promoting 53BP1 nuclear bodies in G1, while RPA protects ssDNA in G2/M to promote 'post-MiDAS' repair. However, the data presented in Figure 4 that is used to support this model is not convincing. The authors have treated cells with nocodazole and EdU simultaneously, used QIBC to identify G1 cells, and then analyzed EdU incorporation at telomeres in this G1 population. Given that EdU was supplemented during mitosis, these EdU foci most likely represent MiDAS events, not DNA synthesis events in G1. These experiments should be conducted where EdU is only supplemented during G1 to confirm post-mitotic repair.

Minor Points

1. The isolation and zooming in on the G1 subpopulation in the QIBC panels is unnecessary.
2. The representative image in 1D should also highlight Cyclin A expression
3. Extended Figure 2c should be more clearly described in the text. This is just a control to demonstrate that the ATMi works, but is just not specifically driving replication stress?
4. The prevalence of TelC and TelG foci in G1 should be quantified and statistically represented.
5. The APB analysis in G1 should be in a single cell stained for RPA, TRF2/Telomeric ssDNA, and PML
6. The increase in RPA foci following FANCM knockdown should be statistically analyzed and colocalization with TRF2 in G1 should be quantified.
7. Extended Data Figure 9D should demonstrate EdU incorporation events at telomeres
8. Both T98G and HT-29 cancer cell lines are known to have an increase in ssDNA in response to replication stress, but not at telomeres. They will have an increase in RPA foci, but unlikely at telomeres given that they represent a non-ALT cell line.
9. Do cells with increased RPA arrest in G1 like has been shown for some 53BP1 events?
10. Do 53BP1 foci in this study lead to post-mitotic repair? Previous publications have demonstrated that some of these events lead to the recruitment of RAD52 to promote BIR?

Reviewer #3 (Remarks to the Author):

In this manuscript, Lezaja et al., discover a new class of DNA lesions inherited through the cell cycle that are shielded by the protein RPA and represent single stranded gaps at telomeres of ALT cells generated at the previous replication cycle. The authors find that these lesions use similar mechanism for their repair to the mitotic DNA synthesis BIR but they are repaired in G1. This is a very carefully executed study that lead to an interesting discovery. The authors use QIBC a method that allows them to temporally and spatially resolve RPA foci. This method is very robust

and allows to analyze a big number of cells and resolve the cell cycle stages in the absence of synchronizing agents that usually cause further DNA damage. Their observation challenges the dogma that the only breaks inherited by the previous cell cycle are shielded by 53BP1 and correspond to fragile sites. For this reason I believe that the manuscript is a good candidate for the journal and I recommend some experiments that would strengthen the conclusions.

1. It was shown that mitotic DNA synthesis requires RAD52. Is it the case for post MIDAs? the authors could make use of RAD52i that seems to work well in several studies.
2. The authors claim that the EU incorporation in RPA positive foci depends possibly on poldelta. To claim this directly they need to deplete poldelta by siRNA and not only use APH and HU to inhibit polymerases.
3. It was shown before that ALT telomeres cluster in PML bodies and their clustering was dependent on RAD51. Interestingly the authors find RPA G1 foci on telomeres at PML bodies even though recombination was shown to be inhibited in G1. The authors could inhibit RAD51 and ask whether the RPA foci do not any more colocalize with PML and whether RAD51 is involved in the repair of the gaps by a recombination mechanism.
4. The authors show clearly that this process takes place in G1. Nevertheless, is there a fraction of gaps at ALT telomeres that starts to get repaired by MIDAS in mitosis and the unrepaired fraction is shielded by RPA in G1?

RPA shields inherited DNA lesions and primes fragile telomeres for post-mitotic DNA synthesis

We would like to thank the reviewers for having taken the time to carefully assess our manuscript and for providing helpful and constructive feedback.

It was very reassuring to read that reviewer #1 considered our work an important contribution to the field and that all experiments in our initial submission were carefully controlled, that reviewer #2 considered the topic we addressed of broad interest to the scientific community, and that reviewer #3 considered our discoveries and conclusions interesting and founded on robust methodology and carefully executed experiments.

Based on insightful and encouraging comments from the reviewers we performed additional experiments and clarified potentially misleading sections in the manuscript text to improve our study further.

Point-by-point response to the reviewers' comments:

Reviewer #1 (Remarks to the Author):

In the manuscript entitled „RPA shields inherited DNA lesions and primes fragile telomeres for post-mitotic DNA synthesis”, Lezaja et al, provide evidence that RPA foci which can be visualized in the G1 phase of the cell cycle are largely due to single stranded gaps at telomeres. They go on to demonstrate that these gaps are due to replication problems that have arisen in the previous cell cycle. Importantly, the RPA foci are distinct from 53BP1 foci. The RPA foci co-localize with ssDNA. Interestingly, in the 4 cell lines that were investigated, it appears that only cells employing ALT are prone to increased telomeric RPA foci in G1. The G1 ssDNA RPA foci co-localize with PML bodies, where it is presumed that ALT telomeres get elongated. By increasing replication stress, either through aphidicolin treatment or the depletion of FANCM, the number of G1 RPA foci could be increased. Finally, the authors demonstrate that the presence of RPA at telomeres is important to prevent the association of 53BP1. In a model the authors present a model that suggests that recombination intermediates in ALT cells get passed onto G1 where they serve to protect ssDNA and promote priming for extension. In general, I find that this is an important contribution to the field and I recommend that it is published in Nat. Comm. I do have a few concerns and suggestions that I think could be addressed before publication occurs. I feel that all experiments have been carefully controlled and that the appropriate statistical tests were applied.

We would like to thank the reviewer for the evaluation and for recommending our study for publication in Nature Communications.

Suggestions:

1. Since only 4 cell lines were employed, I find it difficult to conclude that this only occurs in ALT cell lines. The authors could try expressing telomerase in the ALT cells and see if the RPA foci diminishes. Alternatively, shAsf1 in HeLa cells with long telomeres could be employed to induce isogenic AL

We thank the reviewer for this suggestion. In the revised manuscript, we included two additional cellular systems to compare ALT vs. non-ALT. First, we depleted ASF1 in HeLa cells to induce ALT features as recommended by the reviewer, and indeed observed that this led to an increase in RPA foci in G1 (Extended Data Figure 8h-j). Second, we compared ALT-positive SW26 cells to ALT-negative SW39 cells, both derived from the same origin, i.e. IMR90 cells. These experiments were suggested by reviewer #2, and also here we observed more pronounced RPA foci formation in G1 in the ALT-positive cells (Figure 3f,g). We would like to emphasize, however, that the observed phenotype, i.e. RPA foci in G1, can also be seen in non-ALT cells when they experience high levels

of replication stress (HeLa, RPE-1, SW39), and that ALT-positive cell lines of different cellular origin show some degree of variation in the frequency of RPA foci in G1. Accordingly, we conclude that while cells using ALT are particularly prone to form heritable RPA-marked lesions, this process is not exclusive to ALT-positive cells. We have adjusted the text to better reflect the preferential occurrence in ALT, rather than making a black-and-white discrimination (page 12 and pages 19/20).

2. In the experimental set up shown in figure 5, are there more end-to-end fusions seen in metaphase spreads, this would be expected, but it would be nice to see.

This is an interesting point. Based on this comment, we performed the suggested experiments to score end-to-end fusions in metaphase spreads. However, we see two complications associated with this type of experiment. First, it typically takes at least 3-4 days to observe a significant increase in telomere fusion events¹⁻³, and consistently we only observed elevated fusion when TRF2 was depleted for 5 days but not when TRF2 was depleted for 20h. In line, we did not observe a significant increase in telomere fusions after 20h of RPA depletion, although there was a moderate trend towards more fusion events (Figure R1). Second, performing an RPA depletion in G1 (as we did for Figure 5 in our initial submission) and analysing metaphase spreads afterwards requires progression of cells through S-phase. Given that RPA plays an essential role during DNA replication in S-phase, the results would not allow conclusions about the protective role of RPA at telomeres in G1. We therefore provide the metaphase spread data for the reviewers only.

Figure R1. U-2 OS cells were arrested in G2/M by nocodazole treatment for 16h, a mitotic shake-off was performed and cells were released in fresh medium for 4h, allowing them to pass through mitosis. RPA depletion was then performed for 20h and TRF2 depletion for 20h and 5 days. Afterwards cells were treated with colcemid (0,1ug/ml) for 1h and metaphase spread analysis was performed. 50 spreads were quantified per sample. Individual values and means \pm SD are shown; Mann-Whitney-U test was performed.

3. In the discussion the authors speculate about the source of replication stress in ALT cells stemming from TERRA. This seems reasonable given the phenotypes seen with siFANCM, which reduced R-loops at telomeres. Does the overexpression of RNase H1 reduce RPA foci in G1, this would be easy to test and would hint towards the source of the G1 foci?

4. Building on point 3. Does the depletion of RNase H1 increase the RPA foci?

We agree with the reviewer that the role of R-loops deserves further attention. As the reviewer suggested, we depleted RNaseH1 in two ALT-positive cell lines, U-2 OS and GM847, and in both

cases observed increased RPA foci in G1 (Extended Data Figure 8k-n). We think that this is an important addition to the manuscript, as it provides further support for R-loops as source of replication stress at telomeres, consistent with published data.

The opposite experiment, overexpression of RNaseH1 to reduce RPA foci in G1, yielded inconclusive results. We overexpressed RNaseH1 wild-type and a catalytically inactive mutant in two systems: using doxycycline-inducible GFP-RNaseH1 cells lines⁴ with varying induction periods, and by performing transient overexpression of mCherry-RNaseH1 for 24h to 48h. In both systems we controlled RNaseH1 expression levels by quantitative image-based cytometry (QIBC). Despite extensive experimental attempts, a reduction of replication stress-induced RPA foci in G1 was not observed as a consistent trend in either of the systems. Previous work by the Azzalin group has shown that RNaseH1 overexpression on one hand promotes telomere replication but on the other hand also impairs HR-mediated telomere maintenance⁵. Furthermore, R-loops have been implicated in the activation of RAD52-mediated BIR at telomeres⁶. The levels of TERRA R-loops are probably tightly controlled, and both having too much as well as having too little may compromise telomere maintenance. For these reasons, compared to RNaseH1 overexpression, the experiments with RNaseH1 knockdown included in the revised version of our manuscript are more informative in the context of our study.

5. Finally, if the recombination intermediates are the source of the RPA foci then depletion of RAD52 should also reduce them.

This was an excellent suggestion, shared also by reviewer #3. In the revised manuscript we provide evidence that both knockdown of RAD52 as well as knockout of RAD52 significantly reduces RPA foci in G1 (Figure 5f,g and Extended Data Figures 9d,e & 10c-e). We would like to thank the reviewer for this insightful comment and for the interest in our study.

Reviewer #2 (Remarks to the Author):

In this manuscript, the authors describe a role for RPA in protecting telomeric DNA lesions that arise from replication stress, to facilitate post-mitotic DNA repair. Given the structural complexities of telomeric DNA, telomeres are prone to replication stress. Moreover, replication stress is exacerbated in cells that rely on the ALT pathway for telomere maintenance. ALT telomeres have been demonstrated to rely on break-induced replication (BIR) to not only mitigate replication stress, but to also promote telomere elongation. Mechanistically, BIR has been demonstrated to occur both in interphase, where it is referred to as break induced telomere synthesis (BiTS), as well as metaphase, where it is referred to as mitotic DNA synthesis (MiDAS). Here, the authors describe another type of DNA repair mechanism that they refer to as ‘post-MiDAS’, which represents RPA-coated ssDNA lesions that are repaired in G1. The authors present data to suggest that the type of repair that arises at these DNA lesions is dictated in part by the binding of either 53BP1 or RPA proteins. By exacerbating replication stress at ALT telomeres, the authors demonstrate that both proteins bind DNA lesions in G1 independently of one another, but that there is a substantial increase in RPA foci at telomeric DNA suggesting an increase in telomeric ssDNA gaps. The authors conclude that RPA and 53BP1 may dictate the type of repair that occurs at these heritable lesions with ALT cells being more reliant on RPA.

Fully defining the events that lead to repair of DNA lesions at telomeres is critically important to the understanding of telomere maintenance in ALT and would be of broad interest to the scientific community. However, the data presented in the manuscript almost entirely lack statistical rigor and do not provide enough evidence to support the conclusions of the manuscript.

We were glad to read that this reviewer considers it critically important and of broad interest to define the cellular processes involved in telomere maintenance and ALT. We have carefully revised our manuscript and clarified potentially misleading sections, and we explain all new additions and changes to the manuscript in our point-by-point response to the comments made by the reviewer. We

are confident that the revised manuscript provides conclusive evidence on heritable RPA-marked telomere lesions and telomeric DNA synthesis in G1.

Major Points

1. The terminology the authors use to describe the repair mechanism at ALT telomeres is confusing. MiDAS refers to DNA synthesis events that arise during mitosis. Therefore, post-MiDAS would imply an event that happens after mitotic DNA synthesis. This leads to the confusion. Are the authors suggesting that ssDNA gaps that arise as a byproduct of MiDAS are then further repaired in G1 by BIR? Or are the authors suggesting that RPA coated ssDNA gaps generated during S-phase traverse mitosis and are then repaired in G1. The latter is more consistent with the tone of the manuscript suggesting that the reference should be to post-mitotic repair, not post-MiDAS.

Indeed, MiDAS (“mitotic DNA synthesis”) refers to DNA synthesis events that arise during mitosis. Accordingly, we use the term post-MiDAS (“post-mitotic DNA synthesis”) to refer to DNA synthesis events that occur *after* mitosis. In this sense, no causality is implied, merely a difference in timing, i.e. when the DNA synthesis occurs, and a degree of similarity between the two processes is indicated. Both, MiDAS and post-MiDAS, are processes that deal with unresolved replication intermediates originating from the previous interphase, and both are induced by replication stress. Moreover, MiDAS is frequently referred to as “repair synthesis”⁷⁻⁹; likewise post-MiDAS could be considered “repair synthesis” as well. We see MiDAS as a backup pathway for replication intermediates from S/G2, and likewise we consider post-MiDAS as a backup pathway for intermediates from S/G2 and mitosis. Given the similarities between MiDAS and post-MiDAS we prefer to keep the term, but we define and explain it better in the revised manuscript (page 14/15 lines 294-305, discussion page 19, figure legend for Fig. 7).

2. The entire manuscript is based on the use of U2OS and GM847 cell lines. These cell lines both have inherently low levels of 53BP1 foci (less than 10% of cells have greater than 10 foci), the authors should use an ALT cell line with higher 53BP1 foci to see if this general trend is consistent or unique to this subset of cells. This discrepancy in 53BP1 foci is noted in the literature, but the ‘why’ is still unclear. SAOS2 and SKLU1 are two ALT cell lines that are commercially available and approximately 70% of these cells have greater than 10 53BP1 foci per cell suggesting that these cells may in fact respond differently to DNA damage. The authors should use these cell lines in addition to the U2OS and GM847 to see whether the phenotypes are similar. It is possible, maybe even likely, that the results between these groups of ALT cells may be different suggesting that ALT cells come in different ‘flavors’. Analyzing the SW26 (ALT+), SW39 (ALT-), and IMR90 (parental) isogenic cell lines in this assay would also better support this analysis.

In our initial submission we had compared U-2 OS and GM847 to HeLa and RPE-1 cells. As suggested by the reviewer, we obtained ALT-positive SW26 and ALT-negative SW39 cells from the same origin and quantified RPA foci in G1 cells with and without replication stress. Consistent with our conclusion that ALT cells are more prone to form RPA-marked lesions in G1 compared to ALT-negative cells, SW26 cells showed more RPA foci than SW39 cells, and were more similar to U-2 OS and GM847 as compared to HeLa and RPE-1 (new Figure 3f,g and Extended Data Figure 8f,g).

We also obtained SAOS-2 and SK-LU-1 cells and performed similar analyses, quantifying RPA and 53BP1 foci by cell cycle-resolved QIBC (Figure R2). Both cell lines had high 53BP1 foci counts, but few discernible RPA foci. Both cell lines did not markedly respond to replication stress treatments, and neither showed a strong increase in 53BP1 and RPA foci in G1. These results suggest, as anticipated by the reviewer, that ALT cells indeed come in different ‘flavors’. Importantly, however, when we checked the few RPA foci in G1 that we could observe in SAOS-2 cells, they co-localised with TRF2 and showed EdU incorporation, marking DNA synthesis in G1 (Figure R2i). Thus, the process of post-MiDAS that we describe in our manuscript is conserved in different ALT-positive cell lines (Figure 3, Extended Data Figure 3, Figure 5, Figure R2), but can also be induced in ALT-negative cells by replication stress (Figure 3, Extended Data Figure 3), and the way ALT-positive

cells deal with replication stress varies, possibly depending on their replication timing program and their cell cycle checkpoint control. In our revised manuscript we discuss these points more carefully, e.g. on pages 11/12 and in the discussion on pages 18-20.

Figure R2. (a,b) Asynchronously growing U-2 OS cells were treated as indicated and stained for EdU, RPA and 53BP1. Cell cycle staging was performed based on DAPI and EdU by QIBC. Cell cycle-resolved scatter plots are depicted with RPA **(a)** and 53BP1 **(b)** foci on the y-axis and DNA content on the x-axis. **(c,d)** Asynchronously growing GM847 cells were treated as indicated and stained for EdU and RPA **(c)** and EdU and 53BP1 **(d)**. Cell cycle-resolved scatter plots are depicted with RPA and 53BP1 foci on the y-axis respectively and DNA content on the x-axis. **(e,f)** Asynchronously growing SK-LU-1 cells were treated as indicated and stained for EdU, RPA and 53BP1. Cell cycle-resolved scatter plots are depicted with RPA **(e)** and 53BP1 **(f)** foci on the y-axis and DNA content on the x-axis. **(g,h)** Asynchronously growing SAOS-2 cells were treated as indicated and stained for EdU, RPA and 53BP1. Cell cycle-resolved scatter plots are depicted with RPA **(g)** and 53BP1 **(h)** foci on the y-axis and DNA content on the x-axis. **(i)** Example image of SAOS-2 cells with an RPA focus in G1 co-localization with TRF2 and EdU, marking a site of telomere DNA synthesis in G1.

3. The lack of statistical rigor throughout the manuscript is surprising. All of the data should be analyzed using the proper methods for statistical analysis. In addition, the analysis of such few cells (20 cells in some experiment) is insufficient.

We acknowledge this reviewer's concern. We note that reviewers #1 and #3 seemed quite supportive of the data and the way they were analysed (reviewer #1: "all experiments have been carefully controlled and that the appropriate statistical tests were applied"; reviewer #3: "This is a very carefully executed study that lead to an interesting discovery. The authors use QIBC a method that allows them to temporally and spatially resolve RPA foci. This method is very robust and allows to analyse a big number of cells and resolve the cell cycle stages in the absence of synchronizing agents that usually cause further DNA damage."). Furthermore, very similar cell cycle-resolved single cell and cell population analyses have been performed in previous publications, e.g. Somyajit et al. Science 2017; Saldivar et al. Science 2018; Sedlackova et al. Nature 2020. While in most cases we have analysed several thousand cells per condition (see accompanying Source Data File), for certain types of experiments it is difficult to reach similarly high cell numbers. Also here, ample examples exist in the published literature where few cells (as little as 3-10 cells per condition) have been analysed (e.g. Li et al. Nature 2020; Rey et al. Nature Cell Biology 2020). We nevertheless increased the cell number as much as possible where we had to manually quantify co-localization events in G1 cells (reaching 72 cells in G1 per condition in Figure 6 and 270 cells in G1 per condition in Figure 2), and included statistics as requested (Fig. 1c, 2b-d, 3b-g, 4c, 5b,f,g, 6b,c, and Ext Data Fig. 4a, 8h,i,k,m, 9b,c,d, 10b,c,e).

4. The data in in Figure 1b suggest that upon replication stress the number of RPA foci per nucleus ranges from 0 foci to 20. The graph in Figure 1c averages those foci and determines that on average a G1 nucleus will have 1 foci per cell +/- SD of 0.5. This analysis is not consistent with the data presented by QIBC in Figure 1b, Extended Data Figure 1d, and Figure 2b. In fact, the number of RPA foci vary widely between these figures, this discrepancy is concerning.

We think that there might have been a misunderstanding in the results interpretation, which led to the impression of discrepancy: The QIBC data in Figure 1b displays the range of RPA foci across the cell cycle that can be observed. While this range is indeed from 0 to around 20 foci after APH+ATRI treatment, it does not follow a Gauss distribution, which explains the lower averages in Figure 1c. RPA foci numbers within a G1 population naturally vary in such experiments with asynchronous cells exposed to replication stress treatments, depending on where individual cells were in the cell cycle when they were first exposed to the treatment. Moreover, IF experiments such as the ones performed by QIBC in our study are very well suited for revealing trends and allowing comparisons between conditions, but have inherent limitations when it comes to absolute quantifications of signal intensities and derived parameters (including foci counts) across different experiments and different IF stainings. We nevertheless extended and corroborated our analyses and improved the data visualization (Fig. 1, Fig. 2), and, although some variation between experiments and coverslips cannot be avoided, we observed very consistent trends of RPA foci induction in G1 by replication stress treatments (Figure R3).

Figure R3. (a-d) Asynchronously growing U-2 OS cells were treated as indicated and stained for EdU and RPA in quadruplicates. Cell cycle staging was performed based on DAPI and EdU by QIBC. RPA foci numbers per G1 nucleus are depicted for each treatment and set. Individual values and means \pm SD are shown; Mann-Whitney-U test was performed.

5. The RPA70 foci formation in G1 (Figure 1B) appear to be far more robust than the RPA32 foci formation (Extended Data Figure 1E) in G1 suggesting a discrepancy in the data. Analyzing both RPA subunits side-by-side in QIBC with statistical analysis would help to clarify whether this is actually significant. The authors need to clarify.

RPA32 and RPA70 foci detection depends on specific antibodies, which naturally differ in their target epitope affinity. Absolute quantification and comparison of the signals are therefore not possible. We nevertheless repeated these experiments and compared RPA32 and RPA70 foci detection side-by-side (in the same cells co-stained with both antibodies) by QIBC. Despite the limitations mentioned above with regard to comparing different antibodies, there is a remarkable correlation between the two signals (Figure 1b and Extended Data Figure 1e,f and Figure R4).

Figure R4. (a-b) Asynchronously growing U-2 OS cells were treated as indicated and stained for EdU, RPA70 and RPA32. Cell cycle staging was performed based on DAPI and EdU by QIBC. Cell cycle-resolved scatter plots are depicted for RPA70 (a) and RPA32 (b). (c) Correlation between the RPA70 and RPA32 foci signals under conditions from (a) and (b). (d) Linear regression analysis between the RPA70 and RPA32 foci counts for the APH+ATRi condition corresponding to data from (c).

6. If replication stress exacerbates ssDNA formation at telomeres and this is marked by RPA in G1, and RPA foci increase in G1, why don't the percentage of RPA/TRF2 colocalization events increase in Figure 2d and Extended Figure 5D. This argues that the vast majority of RPA foci that accumulate following replication stress, do not arise at telomeric DNA. This directly contradicts the main conclusions from the manuscript and should be clarified.

The percentage of RPA foci that co-localize with TRF2 foci in G1 does not increase, because the direction of the analysis comes from RPA. Conversely, if we quantify the percentage of TRF2 foci that co-localize with RPA foci in G1, one can see the expected increase following replication stress. In other words, more telomeres are marked by RPA after replication stress treatments. These two complementary but not identical analyses, percentage of RPA foci co-localizing with TRF2 foci and percentage of TRF2 foci co-localizing with RPA foci, are fully consistent with our main conclusions, and in order to clarify this and avoid misunderstandings we have included both analyses side-by-side in the revised manuscript (Figure 2c,d).

7. If telomeres have an increase in ssDNA at telomeres following ATRi and APH treatment the authors should analyse this by S1 nuclease treatment combined with TRF analysis.

We would like to thank the reviewer for this excellent suggestion. We performed the recommended experiment, and indeed observed abolishment of RPA foci, including RPA foci in G1, by S1 nuclease treatment, while TRF2 foci remained largely unaffected (Extended Data Figure 4d and Figure R5).

Figure R5. (a) Asynchronously growing U-2 OS cells were treated with APH, ATRi and APH+ATRi as indicated. Prior to fixation cells were permeabilized and incubated without or with S1 nuclease for 15 min as described in the manuscript methods section. Cells were then stained for RPA and TRF2. Depicted are cell cycle-resolved scatter plots of RPA **(a)** and TRF2 **(b)**.

8. The conclusion that RPA foci represent sites of ssDNA gaps is not supported by experimental data. These RPA foci could represent many different types of DNA lesions that contain single-stranded

DNA, to specifically refer to them as ssDNA gaps would require additional experimental evidence. The data demonstrating colocalization between TelC and RPA, and TelG and RPA, could suggest that the probe is binding one strand and RPA is binding the other, which would indicate a bubble structure, not simply ssDNA. Supporting this, the TelG and TelC show strong correlation under native conditions suggesting both strands are present, but in the form of ssDNA. There are too many possibilities to reliably state that the lesions are specifically ssDNA gaps.

Thank you for this comment. While it would arguably require higher spatial resolution for conclusions about telomeric DNA structures based on the TelC, TelG and RPA co-localization data, we agree with the reviewer that the RPA foci detected in our study could represent different types of ssDNA-containing lesions. We therefore omitted the reference to ssDNA gaps in the revised manuscript and replaced it by ssDNA lesions throughout the text.

9. The authors present a model in Figure 6 to suggest that 53BP1 protects fragile sites in G2/M promoting 53BP1 nuclear bodies in G1, while RPA protects ssDNA in G2/M to promote ‘post-MiDAS’ repair. However, the data presented in Figure 4 that is used to support this model is not convincing. The authors have treated cells with nocodazole and EdU simultaneously, used QIBC to identify G1 cells, and then analysed EdU incorporation at telomeres in this G1 population. Given that EdU was supplemented during mitosis, these EdU foci most likely represent MiDAS events, not DNA synthesis events in G1. These experiments should be conducted where EdU is only supplemented during G1 to confirm post-mitotic repair.

The experiment was specifically designed to **exclude** MiDAS events. Nocodazole was not added to arrest cells in G2/M and then provide EdU in mitosis, but it was instead used on asynchronously growing cells to **prevent** mitotic cells from entering G1. We apologise for the misunderstanding and have explained the experimental setup better in the manuscript text and in the scheme that accompanies the figure (page 13/14, Figure 4f). The short (2h) nocodazole treatment blocks cells in mitosis, as expected (Figure R6), indicating that MiDAS events can indeed be excluded by our experimental setup.

Figure R6. Asynchronously growing U-2 OS cells were treated with nocodazole (50ng/ml) for 2h. Cells were then stained for pH3S10 in order to detect mitotic cells. QIBC-derived cell cycle resolved pH3S10 profiles are shown. Note the increase in mitotic cells marked in red upon the 2h nocodazole treatment. In the manuscript, EdU was provided for the last 1h, in the presence of nocodazole, and cells were then fixed and EdU foci were analysed in G1 cells.

Minor Points

1. The isolation and zooming in on the G1 subpopulation in the QIBC panels is unnecessary.

We adjusted and improved the visualization of the G1 subpopulations in the QIBC panels in order for them to provide more added value.

2. The representative image in 1D should also highlight Cyclin A expression

This panel already includes DAPI (DNA content) and EdU, which is sufficient for G1 identification by QIBC. Moreover, the two antibodies against 53BP1 and RPA cannot easily be combined with a third antibody against Cyclin A in the same staining.

3. Extended Figure 2c should be more clearly described in the text. This is just a control to demonstrate that the ATMi works, but is just not specifically driving replication stress?

Indeed, this is a control to validate the ATM inhibition. We clarified this in the text (page 7) and in the corresponding figure legend.

4. The prevalence of TelC and TelG foci in G1 should be quantified and statistically represented.

Similar to different antibodies with different affinities for their target epitopes, the two FISH probes cannot be compared 1:1. They show very similar trends, however, with increased TelC and TelG foci counts in G1 cells following replication stress treatments (Figure R7).

Figure R7. (a) Asynchronously growing U-2 OS cells were treated as indicated and native (non-denaturing) FISH was performed using telomere-specific Tel C and Tel G probes. Cell cycle staging was performed based on DAPI by QIBC. Quantification of average Tel G foci counts per G1, normalised the untreated sample, are shown in (a) and for Tel C in (b). Individual values and means \pm SD are shown; unpaired t-test was performed.

5. The APB analysis in G1 should be in a single cell stained for RPA, TRF2/Telomeric ssDNA, and PML

Performing indirect immunofluorescence with three different antibodies (RPA, TRF2, PML) on the same cells is limited by the fact that these antibodies are either rabbit or mouse, and combining them would result in cross-labelling. We therefore turned to GFP-RPA cells for the requested experiment, and co-stained TRF2 and PML. The results are included in the revised manuscript (Extended Data Figure 8q).

6. The increase in RPA foci following FANCM knockdown should be statistically analysed and colocalization with TRF2 in G1 should be quantified.

We included the requested analysis in the revised manuscript (Figure 4c and e).

7. Extended Data Figure 9D should demonstrate EdU incorporation events at telomeres.

As requested, we included a new panel to demonstrate EdU incorporation at telomeres (Extended Data Figure 9g).

8. Both T98G and HT-29 cancer cell lines are known to have an increase in ssDNA in response to replication stress, but not at telomeres. They will have an increase in RPA foci, but unlikely at telomeres given that they represent a non-ALT cell line.

We obtained both cell lines and tried to perform QIBC experiments with them. Unfortunately, in our hands both cell lines were not very well suited for IF and showed very few discernible TRF2 foci. For this reason, the suggested co-localization analysis was not possible.

9. Do cells with increased RPA arrest in G1 like has been shown for some 53BP1 events?

This is an interesting point. Unlike 53BP1 nuclear bodies in G1, which are frequently associated with p53 enrichment¹⁰, we find RPA foci in G1 rarely associated with p53. We plan to investigate this more in the future. Furthermore, while 53BP1 nuclear bodies dissolve in G1/S, we could follow RPA foci from G1 through a complete cell cycle. We included an example in the revised manuscript (Extended Data Figure 3d) and briefly discuss this marked difference between 53BP1 and RPA in the corresponding manuscript text.

10. Do 53BP1 foci in this study lead to post-mitotic repair? Previous publications have demonstrated that some of these events lead to the recruitment of RAD52 to promote BIR?

53BP1 foci show little association with RAD52 in G1, and RAD52 was demonstrated to get recruited later in the cell cycle¹¹. Consistently, we observed a low degree of co-localization between 53BP1 and RAD52 in G1 cells (Extended Data Figure 10d,e). Interestingly, however, RPA foci in G1 showed a significantly higher co-localization with RAD52 (Extended Data Figure 10e), indicating a role for RAD52 at RPA-marked lesions in G1. We would like to thank the reviewer for raising this point, which we also extended a bit further based on suggestions by reviewer #3.

Reviewer #3 (Remarks to the Author):

In this manuscript, Lezaja et al., discover a new class of DNA lesions inherited through the cell cycle that are shielded by the protein RPA and represent single stranded gaps at telomeres of ALT cells generated at the previous replication cycle. The authors find that these lesions use similar mechanism for their repair to the mitotic DNA synthesis BIR but they are repaired in G1.

This is a very carefully executed study that lead to an interesting discovery. The authors use QIBC a method that allows them to temporally and spatially resolve RPA foci. This method is very robust and allows to analyse a big number of cells and resolve the cell cycle stages in the absence of synchronizing agents that usually cause further DNA damage. Their observation challenges the dogma that the only breaks inherited by the previous cell cycle are shielded by 53BP1 and correspond to fragile sites. For this reason, I believe that the manuscript is a good candidate for the journal and I recommend some experiments that would strengthen the conclusions.

We would like to thank the reviewer for insightful and constructive suggestions and for considering our study interesting and a strong candidate for publication in Nature Communications. We have done our best to perform the recommended experiments.

1. It was shown that mitotic DNA synthesis requires RAD52. Is it the case for post MIDAs? the authors could make use of RAD52i that seems to work well in several studies.

As the reviewer rightly points out, MiDAS was demonstrated to require RAD52. Interestingly, we observed a significant degree of co-localization between RPA foci in G1 and RAD52 (new Extended Data Figure 10c-e). To validate the specificity of the signals, we had obtained RAD52 knockout cells, which we confirmed by qPCR (Extended Data Figure 10c), and which we then used as control for the RAD52 antibody (Extended Data Figure 10d). The association between RPA and RAD52 in G1 indeed suggests that RAD52 may play a role in post-MiDAS.

As suggested, we then turned to chemical RAD52 inhibitors. Unfortunately, despite many attempts with two different RAD52 inhibitors (AICR and Epigallocatechin, both from Sigma-Aldrich) at various concentrations in both U-2 OS cells and GM847 cells, we did not observe a reduction in RAD52 foci after short (1-6h) inhibitor treatments. Accordingly, we did not observe a change in post-MiDAS as measured by EdU foci in G1 (Figure R8).

Figure R8. (a) Asynchronously growing U-2 OS cells were treated for 6h with RAD52i (epigallocatechin, 20µM, Sigma-Aldrich) and stained for RAD52. RAD52 foci counts per nucleus are depicted. **(b)** Asynchronously growing U-2 OS cells were treated for 24h with APH+ATRi. Nocodazole (50ng/ml) was added for the last 2h to prevent mitotic cells from entering G1. During the same period, cells were exposed to RAD52i (20µM), and for the last 1h EdU was provided. Representative images are shown.

Of note, in published work endogenous RAD52 foci did also not respond to RAD52 inhibitor treatment¹². While upon longer treatment durations and when DNA double-strand breaks are induced ectopically RAD52 inhibition by chemicals may work as a useful means to block RAD52 functions, this strategy did not yield conclusive results in the context of our study. We therefore turned to RAD52 depletion by siRNA and to RAD52 knockout cells. Interestingly, we found that both transient depletion as well as stable knockout of RAD52 resulted in impaired post-MiDAS as measured by reduced EdU foci formation in G1 (Figure 5f,g). Together with the co-localization between RPA and RAD52 in G1 (Extended Data 10e), these results support a function of RAD52 in post-MiDAS, although more work is certainly needed to define its exact role in this process.

2. The authors claim that the EU incorporation in RPA positive foci depends possibly on poldelta. To claim this directly they need to deplete poldelat by siRNA and not only use APH and HU to inhibit polymerases.

We would like to thank the reviewer for this well taken comment. We depleted POLD3, a component of the polymerase delta complex that had been implicated in MiDAS, and surprisingly did not observe loss of EdU foci formation in G1, despite the fact that the depletion was very efficient as judged by qPCR (Figure R9).

Figure R9. (a) Quantification of average EdU foci counts per RPA-positive G1 nucleus in U-2 OS cells transfected with siControl or siPOLD3, treated or not for the last 24h with APH+ATRI. Individual values and means \pm SD are shown. **(b)** Knockdown control by qPCR for POLD3 depletion. Individual values and means \pm SD are shown; unpaired t-test was performed.

Given that this was an unexpected result, we extended our experiments on EdU foci detection in G1. With a higher EdU concentration (100 μ M) and a 1h pulse, EdU foci were readily detectable in otherwise unchallenged U-2 OS cells (i.e. without transfection/knockdown and without replication stress treatments), and their frequency increased with APH, ATRi, and APH+ATRI (new Figure 5a-c). Reassuringly, acute HU and APH, applied together with EdU, abolished EdU foci formation in G1 under these conditions, indicating that these signal originate from DNA synthesis (Extended Data Figure 10b). In light of the POLD3 results (Figure R8) we have revised the manuscript text and emphasize in the discussion that “it will be important to determine in future work whether the post-mitotic DNA synthesis identified in our study uses a similar enzymatic machinery in G1 as the one employed by BIR in mitosis.”

3. It was shown before that ALT telomeres cluster in PML bodies and their clustering was dependent on RAD51. Interestingly the authors find RPA G1 foci on telomeres at PML bodies even though recombination was shown to be inhibited in G1. The authors could inhibit RAD51 and ask whether the RPA foci do not any more colocalize with PML and whether RAD51 is involved in the repair of the gaps by a recombination mechanism.

Also this is an interesting suggestion. We obtained a RAD51 inhibitor, which – different from the RAD52 inhibitors that we tested – reduced foci formation after relatively short treatments (Figure R10a). RPA foci co-localization with PML in G1 cells was only very mildly affected, however (Figure R10b). Consistently, several studies showed that RAD51 has little effect on APB formation¹³⁻¹⁵ and even though RAD51 depletion impaired telomere clustering after DSB induction at telomeres, it did not seem to affect break-induced telomere synthesis. Further, several other publications support a role for RAD52 rather than RAD51 in MiDAS and ALT¹⁶⁻¹⁸, in line with our results on RAD52 and post-MiDAS (Figure 5f,g and Extended Data Figure 10c-e).

Figure R10. (a) Asynchronously growing U-2 OS cells were treated for 6h with RAD51i (B02, 40µM) and stained for RAD51. Cell cycle-resolved scatter plots are depicted with RAD51 foci on the y-axis and DNA content on the x-axis. **(b)** Asynchronously growing U-2 OS cells were treated for 24h with APH+ATRi, with or without RAD51i (40µM) for the last 6h. Co-localization between RPA in PML was manually quantified, 100 cells per condition. Means ± SD are shown; Mann-Whitney-U test was performed.

4. The authors show clearly that this process takes place in G1. Nevertheless, is there a fraction of gaps at ALT telomeres that starts to get repaired by MIDAS in mitosis and the unrepaired fraction is shielded by RPA in G1?

As the reviewer indicates, we indeed consider post-MiDAS as a backup mechanism for MiDAS. While the problems originate from S/G2 (Extended Data Figure 4b), if MiDAS fails or is incomplete (given the short time window from prophase to metaphase during which MiDAS can happen), post-MiDAS may be used. The POLD3 results (Figure R9) suggest that MiDAS is not a prerequisite for post-MiDAS, consistent with this notion. We explain and discuss this point better in the revised manuscript (page 13 and page 19).

Additional references

- 1 Denchi, E. L. & de Lange, T. Protection of telomeres through independent control of ATM and ATR by TRF2 and POT1. *Nature* **448**, 1068-1071, doi:10.1038/nature06065 (2007).
- 2 Vancevska, A. *et al.* SMCHD1 promotes ATM-dependent DNA damage signaling and repair of uncapped telomeres. *EMBO J* **39**, e102668, doi:10.15252/embj.2019102668 (2020).
- 3 Stagno D'Alcontres, M., Mendez-Bermudez, A., Foxon, J. L., Royle, N. J. & Salomoni, P. Lack of TRF2 in ALT cells causes PML-dependent p53 activation and loss of telomeric DNA. *J Cell Biol* **179**, 855-867, doi:10.1083/jcb.200703020 (2007).
- 4 Teloni, F. *et al.* Efficient Pre-mRNA Cleavage Prevents Replication-Stress-Associated Genome Instability. *Mol Cell* **73**, 670-683 e612, doi:10.1016/j.molcel.2018.11.036 (2019).
- 5 Arora, R. *et al.* RNaseH1 regulates TERRA-telomeric DNA hybrids and telomere maintenance in ALT tumour cells. *Nat Commun* **5**, 5220, doi:10.1038/ncomms6220 (2014).
- 6 Tan, J. *et al.* An R-loop-initiated CSB-RAD52-POLD3 pathway suppresses ROS-induced telomeric DNA breaks. *Nucleic Acids Res* **48**, 1285-1300, doi:10.1093/nar/gkz1114 (2020).
- 7 Bhowmick, R., Minocherhomji, S. & Hickson, I. D. RAD52 Facilitates Mitotic DNA Synthesis Following Replication Stress. *Mol Cell* **64**, 1117-1126, doi:10.1016/j.molcel.2016.10.037 (2016).
- 8 Sonnevile, R. *et al.* TRAIP drives replisome disassembly and mitotic DNA repair synthesis at sites of incomplete DNA replication. *Elife* **8**, doi:10.7554/eLife.48686 (2019).
- 9 Franchet, C. & Hoffmann, J. S. When RAD52 Allows Mitosis to Accept Unscheduled DNA Synthesis. *Cancers* **12**, doi:ARTN 26 10.3390/cancers12010026 (2020).
- 10 Kilic, S. *et al.* Phase separation of 53BP1 determines liquid-like behavior of DNA repair compartments. *EMBO J* **38**, e101379, doi:10.15252/embj.2018101379 (2019).
- 11 Spies, J. *et al.* 53BP1 nuclear bodies enforce replication timing at under-replicated DNA to limit heritable DNA damage. *Nat Cell Biol* **21**, 487-497, doi:10.1038/s41556-019-0293-6 (2019).
- 12 Hengel, S. R. *et al.* Small-molecule inhibitors identify the RAD52-ssDNA interaction as critical for recovery from replication stress and for survival of BRCA2 deficient cells. *Elife* **5**, doi:10.7554/eLife.14740 (2016).
- 13 Zhang, J. M., Yadav, T., Ouyang, J., Lan, L. & Zou, L. Alternative Lengthening of Telomeres through Two Distinct Break-Induced Replication Pathways. *Cell Rep* **26**, 955-968 e953, doi:10.1016/j.celrep.2018.12.102 (2019).
- 14 Osterwald, S. *et al.* PML induces compaction, TRF2 depletion and DNA damage signaling at telomeres and promotes their alternative lengthening. *J Cell Sci* **128**, 1887-1900, doi:10.1242/jcs.148296 (2015).
- 15 Potts, P. R. & Yu, H. The SMC5/6 complex maintains telomere length in ALT cancer cells through SUMOylation of telomere-binding proteins. *Nat Struct Mol Biol* **14**, 581-590, doi:10.1038/nsmb1259 (2007).
- 16 Dilley, R. L. *et al.* Break-induced telomere synthesis underlies alternative telomere maintenance. *Nature* **539**, 54-58, doi:10.1038/nature20099 (2016).
- 17 Min, J., Wright, W. E. & Shay, J. W. Alternative Lengthening of Telomeres Mediated by Mitotic DNA Synthesis Engages Break-Induced Replication Processes. *Mol Cell Biol* **37**, doi:10.1128/MCB.00226-17 (2017).
- 18 Min, J., Wright, W. E. & Shay, J. W. Clustered telomeres in phase-separated nuclear condensates engage mitotic DNA synthesis through BLM and RAD52. *Genes Dev* **33**, 814-827, doi:10.1101/gad.324905.119 (2019).

REVIEWER COMMENTS

Reviewer #2 (Remarks to the Author):

In this manuscript, the authors describe a role for RPA in protecting telomeric DNA lesions that arise from replication stress, to facilitate post-mitotic DNA repair. Given the structural complexities of telomeric DNA, telomeres are prone to replication stress. Moreover, replication stress is exacerbated in cells that rely on the ALT pathway for telomere maintenance. ALT telomeres have been demonstrated to rely on break-induced replication (BIR) to not only mitigate replication stress, but to also promote telomere elongation. Mechanistically, BIR has been demonstrated to occur both in interphase, where it is referred to as break induced telomere synthesis (BITS), as well as metaphase, where it is referred to as mitotic DNA synthesis (MiDAS). Here, the authors describe another type of DNA repair mechanism that they refer to as 'post-MiDAS', which represents RPA-coated ssDNA lesions that are repaired relying on DNA synthesis in G1. The authors present data to suggest that the type of repair that arises at these DNA lesions is dictated in part by the binding of either 53BP1 or RPA proteins. By exacerbating replication stress at ALT telomeres, the authors demonstrate that both proteins bind DNA lesions in G1 independently of one another, but that there is a substantial increase in RPA foci at telomeric DNA suggesting an increase in telomeric ssDNA. The authors conclude that RPA and 53BP1 may dictate the type of repair that occurs at these heritable lesions with ALT cells being more reliant on RPA. The authors have been truly responsive in this revised manuscript and performed a number of additional experiments in response to my initial critique. While the author's responses were helpful in clarifying some confusion, they also provided some additional concern about the interpretation of the data and the central conclusions of the manuscript.

1. In the initial review, I felt that there were inconsistencies in the manuscript between the QIBC analysis and the more standard IF analysis. The authors responded that the data do not follow a Gaussian distribution. From a statistical perspective that makes sense, but then the analysis in Figure 1c isn't truly representative of the data as it's not capturing the full range of cells. To me this suggests analyzing more cells to accurately reflect the data. The authors have actually done what appears to be just that in Figure R3. The authors should combine the individual replicates from Figure R3 and use this data in the manuscript? As currently presented in Figure 1, the treatment with aphidicolin alone in the QIBC data is not overly convincing. Perhaps combining the N=4 data in Figure R3 would be more robust with strong statistical support?

2. The data in Figure 1 set the stage for the overarching conclusion that RPA is marking heritable DNA lesions, distinct from 53BP1, that are exacerbated by replication stress. As the data are presented RPA foci do increase in G1, but only following ATR inhibition or the combination of ATRi and Aphidicolin. Aphidicolin alone has no significant and reproducible effect on RPA foci formation in G1 in any of the cell lines except GM847 (maybe U2OS in Figure R3)? However, the authors don't directly address this and instead use the Aphidicolin alone data to support the main conclusion of the manuscript. Were other DNA damaging agents ever used to induce replication stress in these assays (i.e. crosslinking agents, topoisomerase inhibitors, etc) that might offer some clarity? As presented, replication stress alone has a limited role in the heritable lesions observed while the ATRi and ATRi/Aphidicolin treatments demonstrate more consistent results. ATRi alone induces replication stress, but the effects observed here are clearly specific to the ALT cells? Moreover, while Several papers have demonstrated that ALT cells have an increase in RPA foci formation, none have demonstrated that these are specific to G1 cells nor that they are distinct from 53BP1 foci making this finding unique. At face value, the authors are following a logical narrative, but I don't think the data truly support the narrative. I think the authors need to more accurately describe the aphidicolin data and discuss the overall results more directly.

3. The quantifications in Figure 2B, C, and D are a little unclear. The y-axis in Figure 2B refers to 'RPA foci in G1 co-loc with TRF2' is this the average colocalization events with TRF2 per cell? What do the data points represent? Likewise, Figure 2C-D refer to '% foci in G1' is this percentage of cells with foci in G1? It would be more informative and straightforward to graph these as the number of colocalization events per cell. However, the only analysis that reaches significance is the combined ATRi and Aphidicolin condition. The authors support this data with Figure 2E-F, but there is no actual quantification of RPA specifically at ssDNA. Figure 2E shows an increase in telomeric

ssDNA with ATRi and ATRi/Aphidicolin, but not an increase in RPA specifically at telomeric ssDNA. Figure 2F simply provides representative images of RPA at telomeric ssDNA in G1, but no quantification. These data are still not compelling evidence that there is an increase in RPA specifically at telomeric ssDNA in G1. The authors should quantify RPA at telomeric ssDNA or be more direct in the conclusions.

4. The authors should state how many cells are counted in each figure legend (non-QIBC data) and for experiments demonstrating a percentage of positive cells, they should say what constitutes positive, 1 foci? 10 foci?

5. There is still a need for more robust statistical analysis. A Mann–Whitney test was used for statistical analysis of microscopy-derived intensity QIBC data in previous studies and I don't see why that couldn't be implemented here. Likewise, many of the analyses are performed using a student's t-test, given that experiments are often comparing more than two independent variables the authors should use an ANOVA followed by the appropriate post-hoc analysis (i.e. Tukey, Mann-Whitney).

Reviewer #3 (Remarks to the Author):

The authors performed all the suggested experiments which satisfy the concerns. I therefore recommend publication.

RPA shields inherited DNA lesions and primes fragile telomeres for post-mitotic DNA synthesis

Point-by-point response to the reviewers' comments:

Reviewer #2 (Remarks to the Author):

In this manuscript, the authors describe a role for RPA in protecting telomeric DNA lesions that arise from replication stress, to facilitate post-mitotic DNA repair. Given the structural complexities of telomeric DNA, telomeres are prone to replication stress. Moreover, replication stress is exacerbated in cells that rely on the ALT pathway for telomere maintenance. ALT telomeres have been demonstrated to rely on break-induced replication (BIR) to not only mitigate replication stress, but to also promote telomere elongation. Mechanistically, BIR has been demonstrated to occur both in interphase, where it is referred to as break induced telomere synthesis (BiTS), as well as metaphase, where it is referred to as mitotic DNA synthesis (MiDAS). Here, the authors describe another type of DNA repair mechanism that they refer to as 'post-MiDAS', which represents RPA-coated ssDNA lesions that are repaired relying on DNA synthesis in G1. The authors present data to suggest that the type of repair that arises at these DNA lesions is dictated in part by the binding of either 53BP1 or RPA proteins. By exacerbating replication stress at ALT telomeres, the authors demonstrate that both proteins bind DNA lesions in G1 independently of one another, but that there is a substantial increase in RPA foci at telomeric DNA suggesting an increase in telomeric ssDNA. The authors conclude that RPA and 53BP1 may dictate the type of repair that occurs at these heritable lesions with ALT cells being more reliant on RPA. The authors have been truly responsive in this revised manuscript and performed a number of additional experiments in response to my initial critique. While the author's responses were helpful in clarifying some confusion, they also provided some additional concern about the interpretation of the data and the central conclusions of the manuscript.

We would like to thank the reviewer for her/his constructive assessment of our revision. We address and clarify the remaining points below.

1. In the initial review, I felt that there were inconsistencies in the manuscript between the QIBC analysis and the more standard IF analysis. The authors responded that the data do not follow a Gaussian distribution. From a statistical perspective that makes sense, but then the analysis in Figure 1c isn't truly representative of the data as it's not capturing the full range of cells. To me this suggests analyzing more cells to accurately reflect the data. The authors have actually done what appears to be just that in Figure R3. The authors should combine the individual replicates from Figure R3 and use this data in the manuscript? As currently presented in Figure 1, the treatment with aphidicolin alone in the QIBC data is not overly convincing. Perhaps combining the N=4 data in Figure R3 would be more robust with strong statistical support?

The reviewer touches a valid point, namely what is the best representation of the data? We believe that an optimal representation, in the sense of one type of visualisation that is clearly better than all others types, rarely exists. In our manuscript, we therefore use different visualizations to introduce the phenotype of replication stress-inducible RPA foci in G1: Figure 1a and 1b display the cell cycle-resolved single cell

data as they are, without normalisation or averaging. Extended Data Figures 1c and 1d display only the G1 data and, in addition to the non-normalised, non-averaged single cell data in Figure 1a/b, also contain averages and standard deviations to illustrate the variation within the G1 population. Figure 1d contains example images of individual cells to illustrate the same phenotype. Finally, Figure 1c contains averages of replicates for statistical analysis. To this end, we use the average foci counts obtained from hundreds of single G1 cells in each sample, and from these average foci counts calculate new mean values from experimental replicates. Any statistical test based on these means (n=3) is far more conservative than performing statistical tests on single cell data with $n \gg 200$. By definition, the means shown in Figure 1c therefore do not (and cannot) capture the full range of cells (the full range of cells is captured in Figures 1a and 1b, and in Extended Data Figures 1b-f), and analysing more cells would not change this. We clarified this in the figure legend to Figure 1.

The data shown in Figure R3 combined would look like this, with $n > 1000$ per condition. As would be expected, the effects are highly significant ($p < 0.0001$) in all comparisons. The analysis depicted in Figure 1c should be considered more convincing, however, even though the p values are not as low ($p < 0.05$ for the comparison between control and APH). What we have done in addition, based on the second point of the reviewer, is testing increasing amounts of APH at low concentrations (0.2-0.6 μM) to further corroborate the APH effect on RPA foci in G1 (see below).

2. The data in Figure 1 set the stage for the overarching conclusion that RPA is marking heritable DNA lesions, distinct from 53BP1, that are exacerbated by replication stress. As the data are presented RPA foci do increase in G1, but only following ATR inhibition or the combination of ATRi and Aphidicolin. Aphidicolin alone has no significant and reproducible effect on RPA foci formation in G1 in any of the cell lines except GM847 (maybe U2OS in Figure R3)? However, the authors don't directly address this and instead use the Aphidicolin alone data to support the main conclusion of the manuscript. Were other DNA damaging agents ever used to induce replication stress in these assays (i.e. crosslinking agents, topoisomerase inhibitors, etc) that might offer some clarity? As presented, replication stress alone has a limited role in the heritable lesions observed while the ATRi and ATRi/Aphidicolin treatments demonstrate more consistent results. ATRi alone induces replication stress, but the effects observed here are clearly specific to the ALT cells? Moreover, while Several papers have demonstrated that ALT cells have an increase in RPA foci formation, none have demonstrated that these are specific to G1 cells nor that they are distinct from 53BP1 foci making this finding unique. At face value, the authors are following a logical narrative, but I don't think the data truly support the narrative. I think the authors need to more accurately describe the aphidicolin data and discuss the overall results more directly.

APH has a reproducible and significant effect on RPA foci in G1 (Figure 1a-d), although the effect is indeed not as strong as with ATRi or the combination of APH+ATRi. We point this out in the revised manuscript text (page 7) to more accurately describe the APH data as the reviewer suggested. Based on the reviewer's comment, we performed an APH dose-response experiment, using 0.2-0.6 μ M APH. These concentrations are typically used in the field to cause mild replication stress and induce 53BP1 foci in G1. Increasing the APH concentration from 0.2 μ M to 0.3 μ M and to 0.4 μ M led to an increase in RPA foci in G1, which was qualitatively very similar to the increase in 53BP1 foci in G1 (new Extended Data Figures 1e,f). At higher APH concentrations, the number of RPA and 53BP1 foci in G1 decreased, presumably because of delayed progression from S/G2 through mitosis and into G1, which would be expected due to the APH-induced reduction in replication speed and the activation of checkpoint signaling. The newly added APH data corroborate the effect of APH on RPA foci and further strengthen the overarching conclusion that RPA is marking heritable, replication stress-associated DNA lesions.

We also tested other DNA damaging agents to induce replication stress, the crosslinking agent Mitomycin C (MMC), the topoisomerase inhibitor Etoposide, and the PARP inhibitor Olaparib. All three treatments induced 53BP1 foci in G1 cells, and also increased RPA foci in G1 cells. With the new APH dose response experiments showing increasing RPA foci in G1 (new Extended Data Figures 1e,f), and without extensive further characterization of the RPA foci induced by MMC, Etoposide and Olaparib, we prefer to show these data in our point-by-point response:

3. The quantifications in Figure 2B, C, and D are a little unclear. The y-axis in Figure 2B refers to 'RPA foci in G1 co-loc with TRF2' is this the average colocalization events with TRF2 per cell? What do the data points represent? Likewise, Figure 2C-D refer to '% foci in G1' is this percentage of cells with foci in G1? It would be more informative and straightforward to graph these as the number of colocalization events per cell. However, the only analysis that reaches significance is the combined ATRi and Aphidicolin condition. The authors support this data with Figure 2E-F, but there is no actual quantification of RPA specifically at ssDNA. Figure 2E shows an increase in telomeric ssDNA with ATRi and ATRi/Aphidicolin, but not an increase in RPA specifically at telomeric ssDNA. Figure 2F simply provides representative images of RPA at telomeric ssDNA in G1, but no quantification. These data are still

not compelling evidence that there is an increase in RPA specifically at telomeric ssDNA in G1. The authors should quantify RPA at telomeric ssDNA or be more direct in the conclusions.

We relabelled the y-axis in Figure 2b as “Average # RPA foci in G1 co-loc with TRF2” and clarified the figure legends. Figure 2b indeed shows the average number of RPA foci per cell that co-localise with TRF2. This equals the number of co-localisation events. In Figure 2c and 2d we asked what is the percentage of RPA foci in G1 that co-localise with TRF2, and what is the percentage of TRF2 foci in G1 that co-localise with RPA, respectively. Showing such data, in addition to number of co-localisation events per cell (Figure 2b), is important to assess how many of the RPA-marked lesions are at telomeres (around 60-65%) and how many telomeres show signs of inherited replication stress marked by RPA (around 0.5-2%). As requested, we quantified RPA foci in G1 at telomeric ssDNA, the results are shown in Extended Data Figure 7b.

4. The authors should state how many cells are counted in each figure legend (non-QIBC data) and for experiments demonstrating a percentage of positive cells, they should say what constitutes positive, 1 foci? 10 foci?

Where we refer to RPA foci positive G1 cells, we refer to G1 cells in which at least one RPA focus has been detected. We specify this more clearly now in the methods section: “RPA foci positive (i.e. ≥ 1 detected RPA focus) G1 subpopulations”. How many cells were counted is stated in the Reporting Summary: “The following cell numbers were analyzed: >1000 cells per condition for Figures 1a,b; 2e; 3a; 4a,b and Extended Data Figures 1a,b,e,f,g,h; 2a-c; 4a-d; 5a,b; 7b-f; 8d-f,j,l,n; 9a,e; 10a-g. >500 cells per condition for Figures 3e; 4c,e and Extended Data Figures 1c,d,i,j; 3b; 8a,c,g. >250 cells per condition for Figures 1c; 2b-d; 3b,f,g and Extended Data Figures 5e,f. >50 cells per condition for Figures 3c,d; 5b,c,f,g; 6b. >20 cells per replicate condition for Figure 6c”. Depending on the journal preferences, this information could also be copied to each figure legend.

5. There is still a need for more robust statistical analysis. A Mann–Whitney test was used for statistical analysis of microscopy-derived intensity QIBC data in previous studies and I don’t see why that couldn’t be implemented here. Likewise, many of the analyses are performed using a student’s t-test, given that experiments are often comparing more than two independent variables the authors should use an ANOVA followed by the appropriate post-hoc analysis (i.e. Tukey, Mann-Whitney).

As we explained in our response to point 1, a combination of (non-normalised, non-averaged) single cell data plus statistics on absolute or normalised replicate data (by definition this has to be pooled or binned data, losing the single cell information) is in our view ideal and has more value than a Mann–Whitney test on microscopy-derived QIBC data. We performed Mann-Whitney tests on all our QIBC data in this manuscript, and in all cases the test yields highly significant results ($p < 0.0001$), due to the large sample sizes. We are not convinced that incentivising such statistics on single entity data is desirable. Likewise, incentivising averaging of single cell data for statistical analysis of replicate data inevitable leads to loss of information about single cell responses. A combination of both, as we have done in our manuscript, is in our opinion a more accurate way to depict the actual data and allow for evaluation of the

strength and the robustness of the observed phenotypes. We used unpaired t-test for pairwise comparison of average replicate data with the control condition as reference. The test is appropriate for this. Where multiple conditions are compared, in Figures 5f and 5g and in Extended Data Figure 10b, we now use ANOVA with post-hoc Tukey as recommended by the reviewer, and thank the reviewer for this suggestion. All relevant conditions show significant differences. We specify the test used in the correspondingly revised figure legends.

Reviewer #3 (Remarks to the Author):

The authors performed all the suggested experiments which satisfy the concerns. I therefore recommend publication.

We thank the reviewer for assessing our revision and for recommending our study for publication in Nature Communications.

REVIEWERS' COMMENTS

Reviewer #2 (Remarks to the Author):

The authors have addressed all of my comments and were thorough in their revisions. I support publication of the revised manuscript.

Point-by-point response to the reviewers' comments:

We would like to thank the three reviewers once more for their constructive comments and for their interest in our study.

Reviewer #2 (Remarks to the Author):

The authors have addressed all of my comments and were thorough in their revisions. I support publication of the revised manuscript.

We were delighted to read that the reviewer supports publication of our manuscript, thank you.